# GOAL ACHIEVEMENT GUIDED EXPLORATION: MITIGATING PREMATURE CONVERGENCE IN REINFORCEMENT LEARNING

## ABSTRACT

Premature convergence to suboptimal policies remains a significant challenge in reinforcement learning (RL), particularly in tasks with sparse rewards or non-convex reward landscapes. Existing work usually utilizes reward shaping, such as curiosity-based internal rewards, to encourage exploring promising spaces. However, this may inadvertently introduce new local optima and impair the optimization for the actual target reward. To address this issue, we propose Goal Achievement Guided Exploration (GAGE), a novel approach that incorporates an agent's goal achievement as a dynamic criterion for balancing exploration and exploitation. GAGE adaptively adjusts the exploitation level based on the agent's current performance relative to an estimated optimal performance, thereby mitigating premature convergence. Extensive evaluations demonstrate that GAGE substantially improves learning outcomes across various challenging tasks by adapting convergence based on task success. Applicable to both continuous and discrete tasks, GAGE seamlessly integrates into existing RL frameworks, highlighting its potential as a versatile tool for enhancing exploration strategies in RL.

## 1 INTRODUCTION

Properly dealing with the exploration-exploitation trade-off in reinforcement learning (RL) still is a critical challenge (Kaelbling et al., 1996; Ladosz et al., 2022). Constrained by learning time and resources, the agent must balance well between exploring for better policies and exploiting the learned behaviors. There are two prominent challenges in exploration: sparse reward function and local optima. A task with sparse rewards, such as in Montezuma's Revenge, provides insufficient feedback, forcing the agent to search vast areas of the state-action space without clear guidance (Devidze et al., 2022). On the other hand, an environment riddled with local optima may provide the agent with redundant or misleading information and distract it from exploring the actual optimization target. For example, in robot locomotion tasks, where robots are rewarded for saving energy in addition to the main speed reward, agents may focus on optimizing energy consumption but only move slowly. As a result, they may lead to different policy behaviors. Agents trained with sparse rewards might fail to learn any meaningful policy due to exploration difficulty and credit assignment. Whereas agents trained in environments with local optima are more prone to over-exploitation, leading to premature convergence to a suboptimal solution.

Due to RL's trial-and-error nature, local optima can make the learning process unstable. This instability has been reported as a significant obstacle when reproducing and comparing different RL algorithms (Henderson et al., 2018). It is important to distinguish this issue from reward hacking (Amodei et al., 2016), where the agent discovers policies that maximize returns in ways the system designer did not anticipate or desire. We focus on premature convergence, where local optima prevent the agents from optimizing the targeted returns. To effectively solve tasks with local optima, preventing premature convergence during exploration is essential. Several factors contribute to this issue, including the inherent non-convexity of tasks, reward shaping, multi-objectives, and function approximation errors introduced by neural networks in deep RL algorithms.

Many methods have been developed to address the exploration-exploitation trade-off (Ladosz et al., 2022), but not explicitly for premature convergence. One popular approach, $\epsilon$-greedy, employs

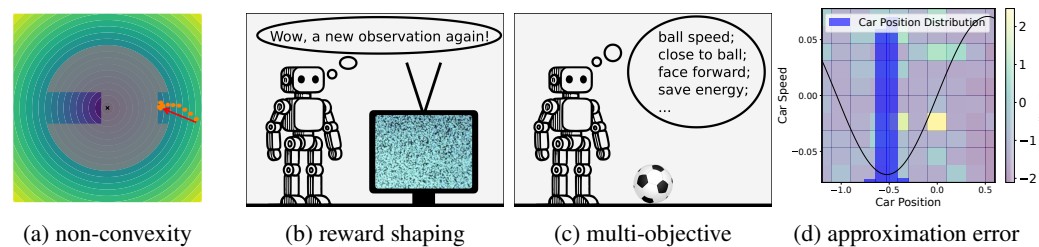

(a) non-convexity     (b) reward shaping     (c) multi-objective     (d) approximation error

Figure 1: Factors for premature convergence in deep reinforcement learning.

a predefined time-decaying parameter $\epsilon$ to decrease exploration gradually. However, finding the optimal schedule is far from trivial, as it can vary depending on the task and is further complicated by the unstable nature of reinforcement learning processes. Other methods, such as Proximal Policy Optimization (PPO)(Schulman et al., 2017) and Soft Actor-Critic (SAC)(Haarnoja et al., 2018), incorporate an entropy-loss component to promote exploration, but this acts only as a soft learning regularizer. Similarly, curiosity-driven intrinsic rewards (Barto, 2013; Pathak et al., 2017) encourage exploration, yet the exploitation process still follows the behavior of the underlying algorithms, like Deep Q-learning (DQN) (Mnih et al., 2013) and PPO, which are prone to premature convergence.

To address the issue of converging to a suboptimal solution prematurely, we propose a novel approach that incorporates an agent's *goal achievement* into its exploration-exploitation strategy. Goal achievement is defined as the ratio of an agent's current policy return to the optimal policy return, excluding the auxiliary rewards for guiding the learning process. Unlike existing approaches, which often overlook this critical aspect of guiding exploration, our approach ensures that exploration continues when the agent's goal achievement is low, thereby preventing early convergence to suboptimal policies. Our method can be applied to both continuous and discrete action spaces. It also has the appealing property of preserving the ranking of different actions while encouraging exploration in discrete tasks, ensuring that the agent does not disrupt valuable policy structures while exploring new possibilities. We demonstrate that this approach can significantly enhance training outcomes across various tasks, potentially improving exploration and overall performance.

To summarize, in this work, we first investigate the various factors that contribute to premature convergence in RL. We analyze existing exploration-exploitation methods and explain why they fail to prevent premature convergence. To solve the problem, we propose **G**oal **A**chievement **G**uided **E**xploration (GAGE), a method that leverages an agent's goal achievement to define an adaptive exploration schedule during training. We evaluate GAGE across multiple challenging tasks, including continuous and discrete action spaces, and compare its performance against popular baseline methods. The results demonstrate that GAGE consistently mitigates premature convergence, especially in complex exploration problems with many local optima. Moreover, GAGE's simplicity and compatibility with a wide range of existing RL algorithms distinguish it as a promising solution for enhancing exploration-exploitation strategies.

## 2   PREMATURE CONVERGENCE AND EXPLORATION TECHNIQUES

Premature convergence is a common issue in optimization and machine learning algorithms like genetic algorithms (Pandey et al., 2014) and reinforcement learning. Despite extensive efforts to enhance exploration efficiency in reinforcement learning (RL) (Thrun, 1992; Auer et al., 2002; Agarwal et al., 2023), agents may still unintentionally converge to local optima due to various factors. In this section, we identify these factors and examine why existing exploration techniques remain prone to this problem.

### 2.1   FACTORS FOR PREMATURE CONVERGENCE

**Non-convexity of tasks**    Non-convexity exists in most real-world tasks and arises from different components, such as the reward function and transition dynamics. It also inherently stems from neural networks, the core part of deep reinforcement learning (DRL). Due to the non-convexity of DRL, sub-optimal solutions can exist even in simple problems. For example, as shown in Fig. 1a, an

agent (orange dot) needs to avoid the grey-colored area and is rewarded more when getting closer to the circle's center. However, due to the non-convexity of the reward landscape, agents without sufficient exploration can get stuck in a local optimal solution (Eckel et al., 2024). In more complex contexts like robotics or traffic management (Xu et al., 2020; Yan et al., 2024a), systems often have many degrees of freedom and complex environmental interactions, making dynamics models non-convex and further complicating optimization.

**Reward shaping** Tasks with sparse rewards and local optima present significant challenges for exploration and credit assignment. To provide agents with dense and informative feedback, previous work has employed reward shaping based on prior knowledge or specific heuristics (Vinyals et al., 2019; Yan et al., 2020). While reward shaping can guide agents toward more valuable regions and accelerate convergence to the optimal policy, designing such rewards is tedious and may introduce local optima (Ng et al., 1999; Gupta et al., 2022; Ma et al., 2024). Agents might focus on auxiliary rewards while neglecting the actual task objectives (Lehman et al., 2020). Curiosity-based intrinsic rewards have become popular for enhancing exploration by rewarding agents for discovering new observations or acquiring new knowledge about the environment (Savinov et al., 2019; Wang et al., 2023). This approach encourages agents to visit diverse states within environments. However, as illustrated in Fig. 1b, agents can become trapped by uncontrolled stochasticity in the system dynamics, a phenomenon known as the Noisy TV problem (Burda et al., 2019).

**Multiple objectives** Many real-world problems involve multiple, sometimes conflicting, objectives that cannot be adequately evaluated using a single metric. For example, as shown in Fig.1c, a robot learning to dribble a football has to optimize factors such as the ball's velocity, energy consumption, distance to the ball, and facing direction. Simultaneously optimizing all these metrics can lead to premature convergence to suboptimal solutions—for instance, the robot might stay close to the ball, face it, and remain stationary to save energy (Yan et al., 2024b). The presence of multiple objectives introduces local optima in the reward landscape, hindering the agent from reaching the global or Pareto optima, depending on the definition of the utility function (Xu et al., 2020; Hayes et al., 2022; Alegre et al., 2023). In this paper, we focus on tasks with linear utility functions that can be addressed using single-objective algorithms rather than exploring Pareto fronts.

**Function approximation error** Neural networks as function approximators enable reinforcement learning (RL) to tackle extremely high-dimensional problems like Go (Schrittwieser et al., 2020). However, they are prone to overfitting (Srivastava et al., 2014), and RL intensifies this issue due to its non-stationarity and biased datasets. As a result, even in simple tasks like Mountain-Car (Moore, 1990), modern algorithms such as Soft Actor-Critic (SAC) can suffer from insufficient exploration (Eberhard et al., 2023), collecting data only around the initial states (see Fig. 1d, where an SAC agent is trained for 1M steps). Due to premature convergence, the learned policy exhibits low entropy even in unvisited states and is thus unable to explore better solutions.

## 2.2 RELATED WORK

Exploration methods for reinforcement learning can be categorized into two groups: undirected and directed (Thrun, 1992). Undirected exploration involves randomly selecting actions based merely on utility estimation. Whereas directed exploration utilizes knowledge of the learning process (Pathak et al., 2017; Burda et al., 2019; Ecoffet et al., 2021) to guide the exploration. In this section, we discuss popular exploration techniques from the two groups.

**Undirected exploration** **1)** The $\epsilon$-greedy strategy, commonly used in value-based algorithms (Mnih et al., 2013; van Hasselt et al., 2016), employs a time-decaying $\epsilon$ to define the probability of selecting either the best action or a random one during training. However, tuning the schedule requires much effort, because many terms can influence the agent's training progress, and the exploration can hardly be defined by the number of iterations. **2)** Some reinforcement learning algorithms are equipped with an entropy loss term (Schulman et al., 2017; Haarnoja et al., 2018) to enhance exploration. However, as a soft regularization for the learning process, it can be insufficient to guide the agent out of local optima. More severely, for discrete actions, the entropy loss can not maintain the distribution shape, i.e., the order of actions' probabilities of the learned policy. **3)** Noise-based techniques inject noise into the observation, action, or parameter space to enhance

policy exploration (Lillicrap et al., 2016; Plappert et al., 2018b). As the magnitude of the noise is controlled by either a time-decaying schedule or learned values, this method has limits similar to those of the previous two.

**Directed exploration** **1)** Curiosity-based methods (Schmidhuber, 1991) are widely employed in hard-exploration environments with sparse rewards. They reward the agent for exploring less visited states. Various approaches have been developed to estimate the novelty of a given state transition, such as the state's visitation number (Tang et al., 2017), the prediction error of a dynamics model (Jarrett et al., 2023), or the information gained through transitions (Nikolov et al., 2019). The Noisy TV problem, as a drawback of curiosity-based methods, has been extensively researched (Savinov et al., 2019; Mavor-Parker et al., 2022; Wan et al., 2023). Yet, a follow-up issue for curiosity-based methods has been neglected: the "Game Console" problem, in which an agent is attracted by a game console with interesting and controllable games instead of a TV showing only noisy images. However, "playing" the game console does not provide the agent with actual rewards. For example, in a maze navigation task, agents may spend much time on a dead-ended long path, gaining curiosity rewards while receiving no actual rewards. When novelty gained from playing games overturns the main objective, curiosity-based methods may prematurely converge to game exploration rather than exploring the optimal solution in the actual world. **2)** Memory-based techniques navigate the agent to promising states as soon as possible through memorizing the visited states (Savinov et al., 2019; Guo et al., 2020; Ecoffet et al., 2021). They reduce the number of frequently visited states near the initial ones, collecting more diverse data and thus mitigating premature convergence by reducing repeated data. However, these methods require high memory, as well as complex state compression and searching processes.

## 3 GOAL ACHIEVEMENT GUIDED EXPLORATION

The learning process should not converge before the agent approaches the maximum possible performance. Therefore, it is natural that the convergence level, reflected by the concentration of the action distribution, is correlated to the goal achievement of the current policy. This section defines goal achievement $g(\pi)$ and explains how it can guide learning convergence.

### 3.1 GOAL ACHIEVEMENT

A reward function is typically composed of several terms, each designed for different purposes. They can be categorized into two groups: goal reward and auxiliary reward. Goal rewards reflect the designer's actual goals, such as winning a game or achieving a target speed, while auxiliary rewards, like curiosity-driven intrinsic rewards, are intended to guide the learning process. Goal achievement of the learning progress should be based on goal rewards, as they directly represent the agent's performance. In contrast, auxiliary rewards do not always align with the actual goals and can lead to suboptimal solutions, as stated in the noisy TV problem. Hence, we exclude these auxiliary rewards when measuring goal achievement.

A task can have more than one goal reward term. For $n_g$ distinct goal rewards, similar to multi-objective algorithms (Xu et al., 2020), we define the goal achievement for each goal as:

$$g_i(\pi) = \frac{\mathbb{E}[V_\pi^{g_i}(s_0)]}{\mathbb{E}[V_*^{g_i}(s_0)]} \,, i \in \{1, \dots, n_g\} \text{ and } 0 \leq g_i(\pi) \leq 1 \tag{1}$$

where $s_0$ represents the initial state, $g_i$ represents the goal achievement of the $i$-th objective among $n_g$ performance metrics, and $V_\pi^{g_i}$ and $V_*^{g_i}$ are the $i$-th components of the vectorized value function for the current policy $\pi$ and the optimal policy $\pi^*$, respectively. We primarily address non-negative goal rewards. For tasks with negative rewards, applying a sigmoid or an offset to the estimated performance can still guarantee that the goal achievement is between 0 and 1. In this work, we define the overall goal achievement of the agent as the minimum goal achievement across all goal reward terms: $g(\pi) = \min(g_i(\pi)) \,, i \in \{1, \dots, n_g\}$. This allows the converged police to optimize jointly all the task-relevant objectives.

In practice, the value function $V_\pi$ often involves significant estimation errors, and computing $V_*$ directly might also be infeasible. Therefore, we approximate $V_\pi$ by using the average rewards from recent rollout trajectories. Determining the optimal performance $V_*$ can often be achieved through

heuristics. For example, in many games, the optimal value of the goal reward $r_{max,t}$ at each step $t$, up to the max episode length $T$, is predefined, such as a fixed value awarded for winning. We can then approximate the goal achievement for a goal reward given the current policy $\pi$ by:

$$g(\pi) \approx \frac{\mathbb{E}_\pi \left[ \sum_{t=0}^T r_t \right]}{\sum_{t=0}^T r_{\max,t}}. \tag{2}$$

Sometimes the explicit knowledge of the reward function or its individual components may not be available. In such cases, the goal achievement can be approximated using the total reward provided by the environment instead of individual goal rewards. When the maximum cumulative reward values are unavailable, the optimal performance can be estimated empirically based on observed performance, as further discussed in Sec. 4.1.

## 3.2 MITIGATING PREMATURE CONVERGENCE VIA ACTION SMOOTHING

To prevent the agent from prematurely converging to local optima and overcommitting to a limited set of actions when goal achievement is low, we apply an action smoothing technique inspired by label-smoothing regularization (Szegedy et al., 2016) for image classification, which reduces overconfidence by smoothing the predicted class distribution. This technique ensures that the agent's action distribution does not collapse into a single action in discrete spaces or into a narrow Gaussian peak in continuous spaces. Below, we discuss how to implement action smoothing in continuous and discrete action spaces.

**Continuous action space**  In continuous action spaces, exploration is typically facilitated by modeling the policy's action distribution as a Gaussian distribution. This approach is used in both stochastic policies like Soft Actor-Critic (SAC) and Proximal Policy Optimization (PPO) (Schulman et al., 2017; Haarnoja et al., 2018), and deterministic policies like Deep Deterministic Policy Gradient (DDPG) (Lillicrap et al., 2016), where the Gaussian distribution serves as additive noise for exploration. The policy learns the mean $\mu(s)$ of the action distribution, modeled as: $p(a \mid s) \sim \mathcal{N}(\mu(s), \sigma^2)$, where the standard deviation $\sigma$ can be controlled via a schedule or learned as a parameter. The standard deviation directly represents the concentration of the action distribution. To prevent premature convergence, we define an adaptive lower bound $\sigma_L(\pi)$ on $\sigma$, which is negatively correlated with the current policy's goal achievement $g(\pi)$:

$$\sigma_L(\pi) = f(g(\pi)). \tag{3}$$

For simplicity, we employ a linear relationship between $\sigma_L$ and the goal achievement $g$, leaving the investigation of other possible functions $f$ for future work:

$$\sigma_L(\pi) = -\sigma_0 g(\pi) + \sigma_0, \tag{4}$$

where $\sigma_0 > 0$ is a hyperparameter controlling the minimum allowed $\sigma$ value when the goal achievement is zero. Agents with a higher $\sigma_0$ require more achievement to concentrate their policies. When $\sigma_0 = 0$, this is equivalent to the original algorithms without GAGE.

**Discrete action space**  In the context of a discrete action space, the probability of each action $a_k$, where $k \in \{1, \dots, K\}$, is usually computed using the softmax function:

$$p(a_k \mid s) = \text{softmax}(z_k) = \exp(z_k) / \textstyle\sum_{i=1}^K \exp(z_i), \tag{5}$$

where $z_i$ is the logit output of the policy network or value network. Existing methods to flatten the action distribution include entropy maximization regularizor (O'Donoghue et al., 2017), label smoothing, and softmax with temperature (Asadi & Littman, 2017). However, the first two of these approaches have significant drawbacks, as illustrated in Fig. 2. For entropy maximization, it controls only the entropy of the whole distribution and does not guarantee that the relative order of the action probabilities remains unchanged, as illustrated in Fig. 2b. Changing this probability order can lead to information loss, potentially impairing the learning process. For label smoothing, it typically mixes the original distribution with a uniform distribution: $p'(a_k \mid s) = (1 - \epsilon)p(a_k \mid s) + \frac{\epsilon}{K}$, where the smoothing parameter $0 \leq \epsilon \leq 1$. Although label smoothing preserves the order of the original probabilities, it can adjust the values inappropriately. For example, actions with the lowest probabilities often lead to penalties or termination states, which the agent should

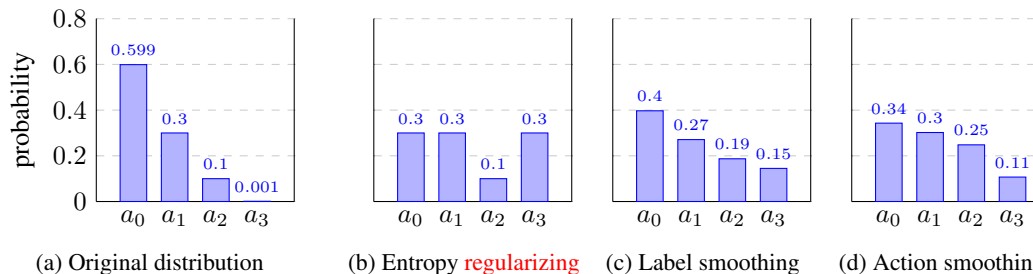

| (a) Original distribution | (b) Entropy regularizing | (c) Label smoothing | (d) Action smoothing |

Figure 2: One categorical distribution and its flattened results using different techniques. The resulting distributions have the same entropy. (a) Original distribution. (b) One possible result through regularization of entropy maximization objective. (c) Label smoothing result calculated with smoothing parameter $\epsilon = 0.58$. (d) Action smoothing result using temperature $\tau = 5.56$.

avoid. However, these actions receive the largest increase (see Fig. 2c), potentially leading to a failure trial. To address these issues, we propose a novel action smoothing method for discrete action spaces using a goal-achievement-guided temperature based on a softmax with temperature. The softmax with temperature function defined as $\mathrm{softmax}(x/\tau)$ modifies the softmax function by introducing a temperature parameter $\tau$. It scales the logits before applying the exponential function, thereby controlling the flatness of the output probability distribution. This method preserves the original probability order of the action distribution, and the largest probability increases occur for actions with middle-valued probabilities rather than the least promising ones (see Fig. 2d). Similar to continuous space, we define an adaptive lower-bound probability for discrete actions. Just as the standard deviation specifies a range of promising continuous actions, we define a range of promising discrete actions, with the number determined by the agent's goal achievement.. More actions are lower-bounded when goal achievement is low. Less are picked when the policy is approaching the goal. Specifically, we adaptively adjust the temperature based on the agent's goal achievement and the original logits, scaling the logit values of all actions to lower-bound the probabilities of the promising actions. The adaptive temperature is calculated as follows:

1. Arrange the policy network output $\{z_1, \ldots, z_K\}$ in descending order to obtain $\{z_{[1]}, \ldots, z_{[K]}\}$ and calculate the differences to the largest value: $z'_k = z_{[k]} - z_{[1]}$, where $z'_1 = 0$ and $z'_k \leq 0$.

2. Determine the number of lower-bounded action probabilities:

$$i_{\mathrm{L}}(g) = -i_0(g(\pi) - 1) + 1 \, , 1 \leq i_{\mathrm{L}}(g) \leq K \tag{6}$$

where the hyperparameter $0 < i_0 \leq K - 1$ controls the number of lower-bounded action probabilities when the goal achievement is zero.

3. Compute the temperature via linear interpolation of the unscaled logits:

$$\tau(g, \boldsymbol{z}') = \eta \max\left(1, \left|z'_{\lfloor i_{\mathrm{L}} \rfloor} + (i_{\mathrm{L}} - \lfloor i_{\mathrm{L}} \rfloor)\left(z'_{\lceil i_{\mathrm{L}} \rceil} - z'_{\lfloor i_{\mathrm{L}} \rfloor}\right)\right|\right), \tag{7}$$

where the hyperparameter $\eta$ controls the lower bound of the top $i_{\mathrm{L}}$ probabilities, and $\max(1, \cdot)$ ensures that the action distribution is only flattened and not sharpened. Note that $i_0, i_{\mathrm{L}} \in \mathbb{R}$ due to interpolation. $\lfloor \rfloor$ is the floor operator and $\lceil \rceil$ is the ceiling operator.

4. Use softmax with temperature to calculate the probabilities $p(a_k \mid s) = \mathrm{softmax}(z'_k/\tau(g, \boldsymbol{z}'))$.

Our approach can also generalize to $\epsilon$-greedy exploration, where the $\epsilon$ can be lower-bounded by an adaptive value based on the goal achievement.

## 4 EXPERIMENTS

This section validates the proposed method by addressing a range of problems characterized by local optima, which often lead to premature convergence in existing reinforcement learning algorithms. First, we apply our approach to solve complex continuous control tasks involving robots with high degrees of freedom. Next, we conduct ablation studies on the hyperparameters to assess the method's

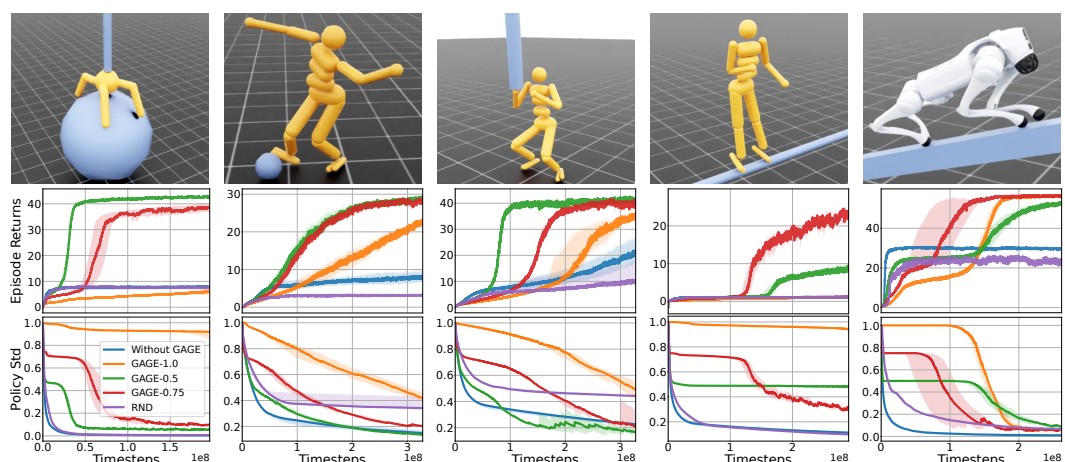

Figure 3: Continuous control experiments. We plot the median over 10 seeds, and the faint area represents the 25% and 75% quantiles, same for Fig. 4. **Top**: tasks from left to right, Ant Acrobatics, Humanoid Dribbling, Humanoid Pole, Humanoid Tightrope, and Dog (Unitree Go2) Balance Beam. **Middle**: Training curves of each method. **Bottom**: Standard deviation of each method $\bar{\sigma}$.

effectiveness in scenarios with unknown optimal goal achievement and evaluate the robustness of the learning process. Finally, we demonstrate the application of GAGE on the "Game Console" problem using the MiniGrid (Chevalier-Boisvert et al., 2023) environments.

## 4.1 CHALLENGING CONTINUOUS CONTROL

To evaluate the effectiveness of our method in preventing premature convergence, we designed five highly challenging continuous control tasks in IsaacLab (Mittal et al., 2023; Yan et al., 2024b) (see Fig. 3). These control tasks jointly include all four factors discussed in Sec. 2.1, featuring non-convex dynamics, complex reward functions, and a hard-exploration nature. The specific environment details are provided in Appendix D. We implemented our approach using Proximal Policy Optimization (PPO), building upon the IsaacLab framework. Since the original action standard deviations are independently learned parameters, we use goal achievement to set a dynamic lower bound for $\sigma$. This is accomplished by applying $\sigma = \sigma_{\mathrm{L}}(\pi)$ whenever it falls below the threshold. The full algorithm is outlined in Appendix A.

**Explore Until Solved** We evaluate our method against two baselines: standard PPO and Random Network Distillation (RND) (Burda et al., 2019). We follow the hyperparameter settings of the original and subsequent work of RND (Yang et al., 2024). The training curves for episode returns and the average $\sigma$ values across all robot joints are shown in Fig. 3. We denote GAGE with $\sigma_0$ value of 0.5 as "GAGE-0.5" in Fig. 3. The proposed method successfully solved all the challenging tasks, whereas the baseline algorithms failed. Notably, PPO without our method is equivalent to GAGE with $\sigma_0 = 0$, and varying $\sigma_0$ can affect the learning process. However, our algorithm remains robust to this hyperparameter over a relatively wide range. The impact of our approach is evident in the plots of $\sigma$. Standard PPO quickly reduces the policy's standard deviation at the start of training, achieving higher rewards by over-exploiting certain reward components, such as energy cost. It continues to decrease the policy's standard deviation even after the target reward plateaus. For instance, the dog robot learns to stand stably on the balance beam and ceases exploration despite having a forward movement target. In contrast, our method keeps exploring and only concentrates the action distribution with increased target rewards. The novelty-based intrinsic rewards of the RND agents slow down the reduction of the policy's standard deviation compared to PPO. However, the additional exploration does not contribute to solving the tasks and, in some cases, even results in worse performance. To better understand the effects of novelty-based exploration in these tasks, we provide additional experimental results with varying intrinsic reward settings in Fig. 7.

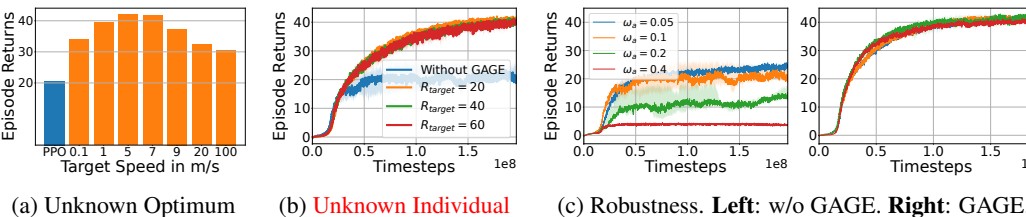

(a) Unknown Optimum       (b) Unknown Individual       (c) Robustness. **Left**: w/o GAGE. **Right**: GAGE.

Figure 4: Ablation Study of GAGE in Humanoid Locomotion Task

**Unknown Optimal Goal**    For some tasks, the optimal performance is well-defined, such as achieving a score of 1 to win in board games. However, for other tasks, like a robot locomotion task, the optimal speed may not be straightforward and is often still under discovery. In the well-known humanoid locomotion task, to achieve higher speed without knowing the optimum, researchers often increase the weighting factors to locomotion speed. However, it can result in unnatural behaviors due to imbalanced speed and action rewards. To address this issue, we conducted experiments to demonstrate how GAGE can explore higher speeds without knowing the maximum or altering the reward weights. We define the goal achievement in the Humanoid task as $g_\pi = v_\pi / v_*$, where $v_\pi$ and $v_*$ represent the robot's current and target speeds, respectively. As shown in Fig. 4a, our method significantly outperforms standard PPO across a wide range of target speeds, from 0.1 to 100 m/s while maintaining natural behavior through balanced reward weights. When the target speed is set below the learned optimal speed ($\sim$ 7 m/s), the GAGE agent is also able to learn the optimal speed. This phenomenon highlights the critical importance of exploration during the initial stages of training. At the very beginning, none of the agents had learned to move forward, i.e. $g(\pi) \approx 0$. As a result, even with small target speeds of 0.1 or 1, the GAGE agent maintained higher standard deviation values compared to standard PPO. Since the optimal locomotion gait for the humanoid robot remains consistent across different speeds, it is crucial for the agent to avoid becoming trapped in a suboptimal policy (gait) early in training. Once the agent learns an effective low-speed gait, gradually increasing locomotion speed with a similar gait requires less exploration. Even when the target speed is set unreasonably higher than the optimal speed, the GAGE agent is able to discover improved performance, which can subsequently be used to refine the estimation of the optimal performance.

**Unknown Individual Rewards**    Sometimes, the agent only receives a total reward from the environment and lacks access to individual reward components. As the optimal cumulative return is usually unknown, we propose to approximate this value based on the final performance of the standard PPO agent. We conducted additional experiments on the humanoid locomotion task, where we performed an ablation study using $1\times$, $2\times$, and $3\times$ of the standard PPO's episode return as the estimated optimal return. The results, depicted in Fig. 4b, show the episodic returns averaged over the last 10 episodes across 5 different seeds. These results demonstrate that GAGE consistently improves performance in this scenario.

**Improved Robustness to Reward Shaping**    Reward shaping is crucial yet challenging in reinforcement learning, as even minor adjustments to the weighting of specific rewards can result in unsuccessful learning. Using the humanoid locomotion task, we demonstrate this issue and the effectiveness of our method in mitigating it. As in the previous experiment, we define goal achievement based on locomotion speed. The reward terms include a penalty for large action values, $\omega_a \|\boldsymbol{a}\|^2$. In the experiments, we kept the weights for other rewards constant while varying $\omega_a$. The baseline agents without our method exhibited performance that was highly sensitive to changes in $\omega_a$, with significant impacts on both final speed and locomotion gait. In contrast, our method enabled the agent to maintain high running speeds and achieve consistently high returns across all the $\omega_a$ values (see Fig. 4c).

## 4.2 "GAME CONSOLE" PROBLEM

We further validate our method in discrete action spaces using the MiniGrid environments (see Appendix D). This benchmark has gained significant attention from reinforcement learning researchers.

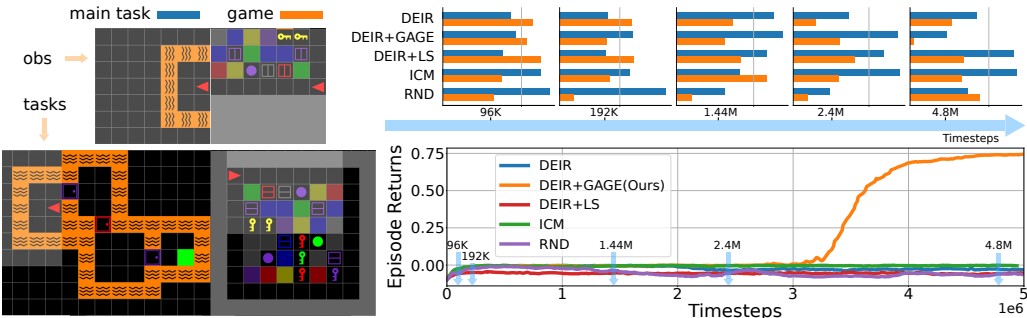

Figure 5: Experiments in MiniGrid. **Left**: The environment consists of two tasks. Their observations are concatenated as the environment observation. The main task, "MultiRoomLava-N4-S5," has six actions. The gaming task has three. **Right up**: We evaluate each agent 10 times and average the number of actions it takes in the main and the gaming tasks. The vertical grey line in each plot represents action number 40. **Right bottom**: Learning curves of different agents.

Numerous methods, particularly curiosity-based approaches, have been developed to successfully solve most tasks, even with the inclusion of Noisy TV as a distraction. Through this experiment, we aim to address two key questions: 1) Will curiosity-based agents be drawn to controllable distractions? 2) If so, can GAGE mitigate such premature convergence? To explore these questions, we introduce the following major modifications to some of the original environments (see Fig. 5):

• Attach a "Game Console" to each environment. Unlike the uncontrollable stochasticity in the Noisy TV problem, we investigate controllable distractions within the environment. In addition to the original MiniGrid task, where the agent learns to navigate and reach specific targets, the agent can simultaneously interact with a parallel gaming task. This gaming task features a larger map with more randomly located objects, providing the agent with novel observations as it navigates. The agent receives an episodic reward solely for reaching the goal in the main task. The observations and actions from both tasks are concatenated for the entire environment.

• Introduce a penalty to the main task. To increase the learning difficulty associated with premature convergence, all the walls in the main task were replaced with lava. If the agent in the main task steps on lava, the environment terminates, and a penalty is applied. In contrast, the gaming task contains no lava, making exploration in the gaming task more appealing due to its safety.

Our method builds on the Discriminative-model-based Episodic Intrinsic Reward (DEIR) (Wan et al., 2023), an intrinsic reward algorithm designed to address the Noisy TV problem in Mini-Grid. We adjust the policy network by incorporating an adaptive softmax temperature, guided by the agent's goal achievement, as described in Sec. 3.2. To evaluate our approach, we compare it against three baseline algorithms: Intrinsic Curiosity Module (ICM) (Pathak et al., 2017), Random Network Distillation (RND) (Burda et al., 2019), and DEIR. Additionally, we conduct an ablation study to investigate the effectiveness of using label smoothing (LS) to guide exploration.

All baseline algorithms fail to solve the tasks. They were distracted by the gaming environment and struggled to learn meaningful policies for the main task. However, their behavior patterns differed during the learning process. The training results for a single seed on the MultiRoomLava-N4S5 environment are shown in Fig. 5, with additional results in Appendix B. Above the training curves, we present the action distributions of different agents during training. Each agent was evaluated 10 times, and the average number of actions taken in the main and gaming tasks was computed. The ICM and RND agents were initially less attracted to the gaming task. At convergence, the ICM agent learned to stay alive, while the RND agent learned to terminate the environment shortly after it started. The DEIR agent initially explored the gaming task extensively. As it became familiar with the game, it gradually shifted its attention to the main task. However, due to the action distribution having already converged to specific actions, it struggled to complete the main task and collect extrinsic rewards. Ultimately, it also learned to terminate the environment, similar to the RND agent.

The agent using label smoothing also fails to solve any of the tasks. Although it maintains high entropy when no external rewards are achieved, its exploration is less effective compared to GAGE agents. This is because the uniform distribution keeps the probabilities of undesired actions, such as those leading to Lava cells (termination), relatively high during training. This behavior is evident in the episode length plots shown in Fig. 8. These results are consistent with our hypothesis.

The agent using our method is initially distracted by the game. However, GAGE ensures a lower bound on the probabilities of promising actions when the goal achievement is zero, allowing the agent to maintain sufficient exploration in the main task. It also learns to avoid the lava, as a few action probabilities are not lower-bounded and can be reduced to low values. After thoroughly exploring the main task, the DEIR+GAGE agent successfully navigates to the goal and begins collecting extrinsic rewards, which are significantly larger than the curiosity-based intrinsic rewards. Eventually, the agent shifts its focus entirely to the main task, abandoning the exploration of the gaming task.

## 5 DISCUSSION

We introduced Goal Achievement Guided Exploration (GAGE), a method aiming to address premature convergence in reinforcement learning (RL). Our approach uses goal achievement as a dynamic factor to guide the agent's exploration, allowing for a better balance between exploration and exploitation. Our experiments demonstrate that GAGE substantially mitigates premature convergence in complex environments by maintaining adequate exploration. Unlike traditional methods such as entropy maximization or curiosity-based exploration, GAGE incorporates an adaptive mechanism that smoothes the action probability distribution based on how well the agent achieves its goal. The strength of GAGE lies in its simplicity and compatibility with existing RL algorithms. It does not require significant architectural changes and can be easily integrated into continuous and discrete action space environments. The flexibility of GAGE makes it applicable to a wide variety of real-world RL problems.

Despite these strengths, GAGE offers aspects for improvement. The current version relies on defining an appropriate goal achievement metric, which might not be straightforward in all tasks. In environments in which the optimal policy or goal is not well understood, the approximation of goal achievement might introduce inaccuracies. Additionally, while GAGE has proven effective in the tested environments, its scalability to more complex, high-dimensional tasks has yet to be explored.

Future research should focus on improving the scalability of GAGE and applying it to more complex, dynamic and multi-objective environments. Investigating non-linear relationships between the goal achievement and the exploration metrics, such as the standard deviation of Gaussian distributions in continuous action spaces, could further enhance the method's adaptability to diverse RL problems.

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

# A   ALGORITHM IMPLEMENTATION

We provide the pseudo code for PPO+GAGE with Gaussian policy in Alg. 1 and action smoothing with categorical policy in Alg. 2.

---

**Algorithm 1** Proximal Policy Optimization (PPO) Algorithm with Gaussian Policy + GAGE

---

1: Initialize policy mean parameters $\theta_0$, value function parameters $\phi_0$, standard deviation $\sigma_0$, and goal achievement $g_0$
2: **for** iteration $k = 0, 1, 2, \ldots$ **do**
3:     Collect set of trajectories $\{(s_t, a_t, r_t, s_{t+1})\}$ by running policy $\pi_{\theta_k}(a_t|s_t) = \mathcal{N}(\mu_{\theta_k}(s_t), \sigma_k^2)$ in the environment
4:     **for** each time step $t$ **do**
5:         Compute advantage estimates $\hat{A}_t$ based on value function $V_{\phi_k}(s_t)$
6:     **end for**
7:     Update the policy by maximizing the PPO-CLIP objective with an added entropy term:

$$\theta_{k+1}, \sigma_{k+1} = \arg\max_{\theta,\sigma} \mathbb{E}_t \Big[ \min\left( \frac{\mathcal{N}(\mu_\theta(s_t), \sigma^2)}{\mathcal{N}(\mu_{\theta_k}(s_t), \sigma_k^2)} \hat{A}_t, \text{ clip}\left( \frac{\mathcal{N}(\mu_\theta(s_t), \sigma^2)}{\mathcal{N}(\mu_{\theta_k}(s_t), \sigma_k^2)}, 1-\epsilon, 1+\epsilon \right) \hat{A}_t \right)$$

$$+ \beta H(\pi_\theta(a_t|s_t)) \Big]$$

    where $\mu_{\theta_k}(s_t)$ is the mean of the Gaussian action distribution, $\sigma_k$ is the standard deviation (separately learned), and $H(\pi_\theta(a_t|s_t))$ is the entropy of the policy, encouraging exploration. The term $\beta$ controls the weight of the entropy regularization.
8:     Update the value function by minimizing the following loss:

$$\phi_{k+1} = \arg\min_\phi \mathbb{E}_t \left[ (V_\phi(s_t) - R_t)^2 \right]$$

9:     Calculate the running mean of $g_k$.
10:     Update the standard deviation parameter $\sigma$ based on the agent's performance:

$$\sigma_{k+1} = \max(\sigma_{k+1}, -\sigma_0 g_k + \sigma_0)$$

11: **end for**

---

**Algorithm 2** Action Smoothing Algorithm

---

**Require:** Network outputs $\{z_1, z_2, \ldots, z_K\}$, goal achievement $g(\pi)$, hyperparameters $\eta, i_0$
**Ensure:** Action probabilities $p(a_k \mid s)$
1: **Order the network outputs** in descending order to obtain $\{z_{[1]}, z_{[2]}, \ldots, z_{[K]}\}$ such that $z_{[1]} \geq z_{[2]} \geq \cdots \geq z_{[K]}$
2: **Compute differences** to the largest value:

$$z'_k = z_{[k]} - z_{[1]}, \quad \text{for } k = 1, 2, \ldots, K \quad (\text{Note: } z'_1 = 0)$$

3: **Decide the number of top actions** based on goal achievement:

$$i_{\text{L}}(g) = -i_0(g(\pi) - 1) + 1$$

4: **Calculate the temperature**:

$$\tau(g, \boldsymbol{z'}) = \eta \max\left( 1, \left| z'_{\lfloor i_{\text{L}} \rfloor} + (i_{\text{L}} - \lfloor i_{\text{L}} \rfloor)\left( z'_{\lceil i_{\text{L}} \rceil} - z'_{\lfloor i_{\text{L}} \rfloor} \right) \right| \right)$$

5: **Compute action probabilities** using softmax with temperature:

$$p(a_k \mid s) = \frac{\exp(-z'_k / \tau(g, \boldsymbol{z'}))}{\sum_{i=1}^K \exp(-z'_i / \tau(g, \boldsymbol{z'}))}, \quad \text{for } k = 1, 2, \ldots, K$$

---

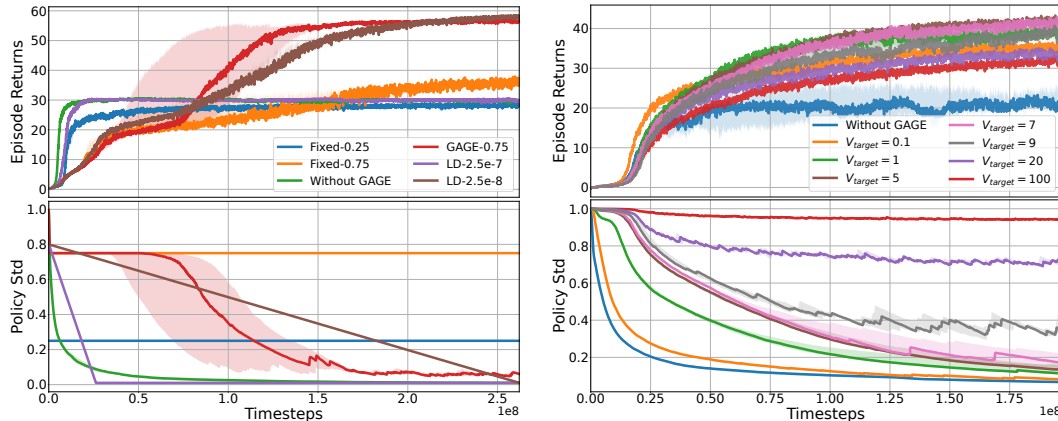

(a) Dog Balance Beam including agents trained with $\sigma$-schedule

(b) Unknown optimal goal on humanoid locomotion task

Figure 6: Additional training results of experiments with continuous action space. We plot the median over 5 and 10 seeds. The faint area represents the 25% and 75% quantiles.

## B ADDITIONAL RESULTS

### B.1 CONTINUOUS ACTION

In this section, we provide more detailed experiment results with continuous action spaces.

**$\sigma$ Schedule** To compare our method with exploration approaches using constant or linearly decreasing standard deviations, we conducted experiments on the Dog Balance Beam task. The agent was trained with constant $\sigma$ values of 0.25 and 0.75, as well as with linearly decreasing schedules ranging from 0.8 to 0.01 over $2.5 \times 10^7$ and $2.5 \times 10^8$ timesteps. As shown in Fig. 6a, only the agent with a linearly decreasing standard deviation similar to the curve discovered by our method achieved performance comparable to GAGE. This result further validates the effectiveness of our approach. Additionally, since tuning a predefined entropy schedule—considering both entropy values and training duration—is highly resource-intensive, our method significantly reduces the workload by introducing an adaptive schedule.

**Unknown Optimal Goal** To provide more detailed insights, we present the learning curves for episode return and policy standard deviation in the humanoid locomotion task with an unknown optimal goal, evaluated across different target velocities, in Fig. 6b.

**Intrinsic Reward Weight** To evaluate the effect of intrinsic rewards in the proposed challenging control tasks, we trained several RND agents using different weight combinations for extrinsic and intrinsic rewards: (2.0, 1.0), (2.0, 0.5), (1.0, 1.0), (1.0, 2.0), and (1.0, 4.0). The weight values (2.0, 1.0) are consistent with those used in the original RND work (Burda et al., 2019) and subsequent research (Yang et al., 2024). Therefore, we also used this ratio for the experiments presented in Fig. 3. As shown in Fig. 7, none of the RND agents succeeded in solving the task. Agents with larger ratios of extrinsic-to-intrinsic weights exhibited learning patterns similar to standard PPO, which does not use intrinsic rewards. As the ratio decreased, the agents focused more on exploring novel states, as indicated by larger standard deviations during training. However, this increased exploration did not contribute to solving the task. Instead, the novelty-based exploration resulted in decreased extrinsic rewards. This phenomenon highlights the distinct focus of our work compared to novelty-based exploration methods. Our work focuses on addressing premature convergence, an issue that is equally

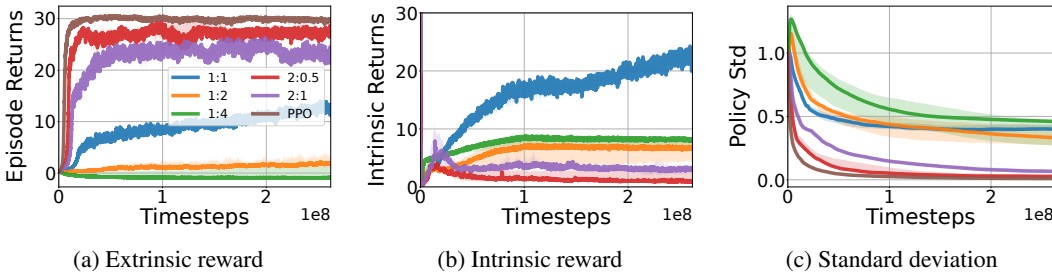

Figure 7: Investigating the effect of novelty-based intrinsic reward to the learning of Dog Balance Beam task. The curves with legend 1:2 represent the agent trained using extrinsic and intrinsic coefficients of (1.0, 2.0).

important but has been largely overlooked until now. In contrast, curiosity-based methods primarily tackle sparse rewards. The difference in focus is also reflected in the existing benchmarks for exploration algorithms. Most environments are designed with sparse rewards and moderate local optima, which can be effectively addressed using novelty-based exploration. For example, environments like Fetch (Plappert et al., 2018a), MiniGrid (Chevalier-Boisvert et al., 2023), AntMaze, and Adroit manipulation tasks (Fu et al., 2020) are "safe," with sparse termination states or penalties distributed across the state space. Agents can easily avoid termination and penalty states while exploring for rewards. In such environments, exploring unseen states is a highly effective strategy. However, novelty-based methods struggle in scenarios with more severe and deeper local optima. For instance, Noisy-TV has been recognized as a major issue for novelty-based methods, even though it only involves local optima introduced by environment stochasticity. The challenges posed by more severe local optima have not yet been fully explored. In this work, we aim to push the boundaries of RL exploration research into environments with more challenging local optima issues. The proposed IsaacLab tasks reflect real-world robot control scenarios where optimal behaviors occupy only a small portion of the state space, while most of the state space leads to penalties such as falling down or wasting energy. This dominant penalizing space creates challenging local optima. In such environments, novelty-based exploration often results in sampling mostly failed trajectories and becoming trapped in local optima. A similar phenomenon is observed in the MiniGrid experiments, where popular novelty-based methods fail to solve tasks with more challenging local optima.

## B.2 DISCRETE ACTION

In this section, we provide more detailed experiment results with discrete action spaces. The learning curves, including episode return, entropy loss, and episode length, are shown in Fig. 8. Our proposed method, combined with DEIR, can successfully solve all the tasks. The baseline algorithms, however, all fail to learn meaningful policies with the introduced "Game Console" problem. Due to the lack of GAGE, their entropy losses quickly increase from around -2.0 to -1.0 or even higher. The concentrated action distribution impairs the agent's search for the main task reward, which requires long-horizon exploration. Our method helps the agent maintain a relatively low entropy loss when the goal achievement is zero, preserving its ability to explore. The action distribution only further sharpens after the agent achieves an improved performance.

Interestingly, the baseline algorithms show different behavior patterns during learning (see Fig. 5). As shown in the plots of episode length, the RND agents quickly learn to terminate the environment by navigating onto lava elements. The ICM agents also learn to terminate at the beginning but then change to the strategy of staying alive and continuing to explore the gaming task. The DEIR agents, however, first focus on exploring the gaming task and then change to the termination policy.

## C ACTION SMOOTHING CALCULATION OF DISCRETE ACTION SPACE

In Fig. 9, we illustrate the calculation of action smoothing with discrete action space for $\eta = 1$ and $i_L = 3$. By ensuring $\tau \geq 1$, our method only flatten and does not sharpen the original distribution. In Fig. 10, we show the theoretical lower-bounded probabilities of different actions after action

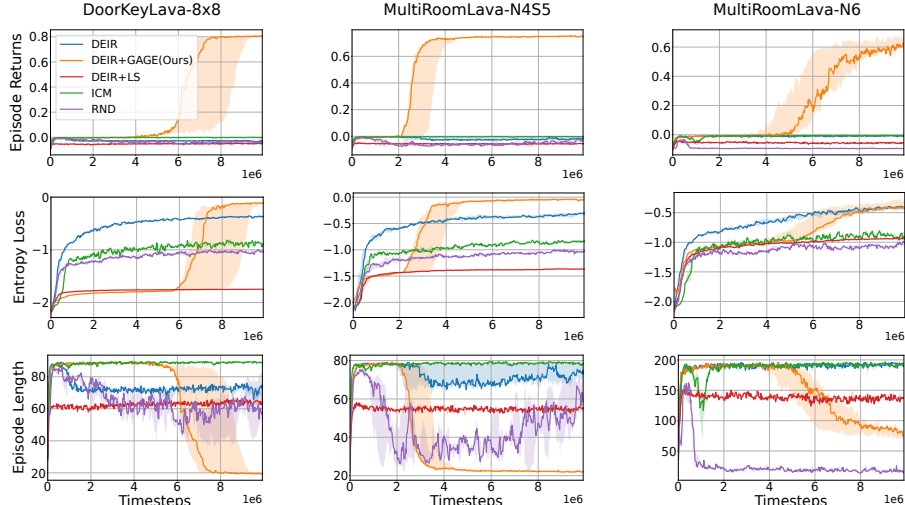

Figure 8: Training results of experiments with discrete action space. We plot the median over 5 seeds, and the faint area represents the 25% and 75% quantiles.

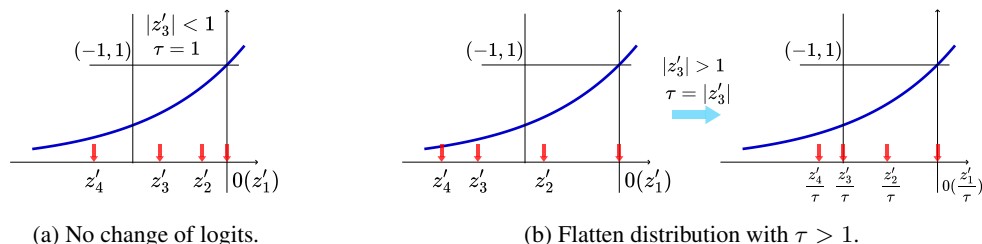

(a) No change of logits.     (b) Flatten distribution with $\tau > 1$.

Figure 9: Softmax temperature calculation with $\eta = 1$ and $i_L = 3$.

smoothing for $\eta = 1$ and $i_L = 3$. As $\tau = |z_3'|$, the probabilities $p(a_1), p(a_2)$ and $p(a_3)$ are all lower-bounded, whereas $p(a_4)$ can asymptotically approach zero. We can calculate the lower bounds for different action probabilities as $p_L(a_1) = 0.37, p_L(a_2) = 0.17, p_L(a_3) = 0.13$.

# D    EXPERIMENTAL DETAILS

## D.1    TASKS SETUP

We build up five challenging continuous control tasks in IsaacLab[1]. Three robots with many degrees of freedom learn to do challenging locomotion or dynamic manipulation behaviors. The robots include a humanoid robot with 21 joints, a dog robot (Unitree Go2) with 12 joints, and an ant robot with 8 joints. The humanoid robot is also employed in the locomotion task to investigate maximum speed and robustness to reward weights. In Table 1, we provide the reward composition of different tasks.

**Humanoid tightrope (HT)**   The humanoid robot learns side walking on a tightrope, i.e., a cylindrical bar with a diameter of only $0.1$m. This is more challenging than walking forward because balancing with two arms stretching to both sides would be more difficult.

**Humanoid dribbling (HD)**   The humanoid robot learns to dribble a football at a high speed ($3.5$m/s). Additionally, the robot gets random commands for turning the target direction for up to $\frac{\pi}{4}$rad.

---

[1]https://isaac-sim.github.io/IsaacLab/

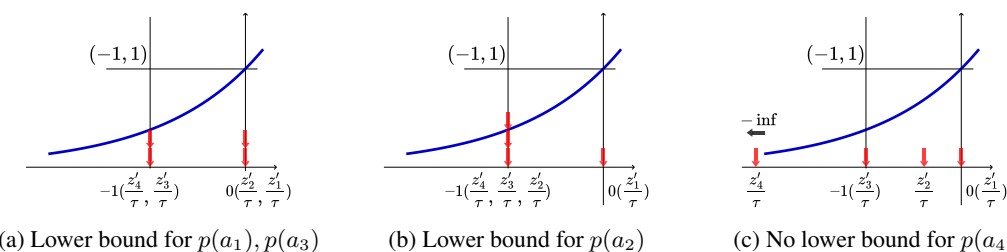

(a) Lower bound for $p(a_1), p(a_3)$     (b) Lower bound for $p(a_2)$     (c) No lower bound for $p(a_4)$

Figure 10: Lower-bounded probability calculation after action smoothing with $\eta = 1$ and $i_L = 3$.

Table 1: Reward weights of continuous control tasks. The rewards and penalties from left to right are for robot locomotion velocity, environment not terminating, robot orientation, robot distance to the manipulated object, large action commands, energy consumption, joint position too close to limitations, robot velocity perpendicular to the desired direction, object velocity perpendicular to the desired direction, joint torque, joint acceleration, and action changing rate. The selected goal reward for goal achievement calculation is marked in green background.

| | **reward** | | | | **penalty** | | | | | | | |
|------|-------|-------|--------|-------------|-----------|-------|-------------------|-------|--------------|------|-------------|-------------|
| | $v_x$ | alive | orient | $d_{\text{obj}}$ | $\|a\|^2$ | $E$ | $\theta_{\text{limit}}$ | $v_y$ | $v_{y,\text{obj}}$ | $T$ | $\ddot{\theta}$ | $\dot{a}$ |
| HT | 0.5 | 1.0 | 1.0 | 0 | 0 | 0.05 | 0.25 | 1.0 | 0 | 0 | 0 | 0 |
| HD | 0.3 | 0.4 | 1.0 | 0.2 | 0.01 | 0.01 | 0.25 | 0 | 0.5 | 0 | 0 | 0 |
| HP | 2.0 | 1.0 | 1.0 | 0 | 0.01 | 0.005 | 0.125 | 0 | 1.0 | 0 | 0 | 0 |
| DB | 1.0 | 1.0 | 1.0 | 0 | 0.005 | 0 | 0 | 1.0 | 0 | 1e-6 | 2.5e-8 | 0.001 |
| AA | 1.0 | 1.0 | 1.0 | 0 | 0.005 | 0.05 | 0.1 | 0 | 1.0 | 0 | 0 | 0 |

**Humanoid pole (HP)** The humanoid robot learns to walk forward while balancing a pole vertically on its right hand. The target walking speed is $0.5\text{m/s}$ and the pole is $2\text{m}$ long.

**Dog balance beam (DB)** The dog robot learns to walk on a balance beam. The beam has a square crosssection with $0.1\text{m}$ side length. Moreover, the balance beam is tilted for $\frac{\pi}{9}\text{rad}$ so that the robot has to climb a slope while balancing.

**Ant acrobatics (AA)** The ant robot with four legs learns to balance a pole vertically on its torso while standing on a ball. The pole has a length of $2\text{m}$. The ball has a diameter of the same value. Moreover, the robot has to learn to roll the ball forward at a target speed of $1\text{m/s}$.

To validate the performance of our method with discrete action space, we build up three MiniGrid[2] environments with the Game Console problem. To validate the effectiveness of our proposed method against local optima, we improve the difficulty of the original MiniGrid environments by 1) changing all the wall elements to lava and 2) attaching a parallel gaming task to the original tasks. In the main tasks with lava walls, the agent learns to navigate to the target to get an episodic reward. The environment terminates once the agent navigates onto a lava element. In the gaming task, the agent can navigate to different positions but acquires neither rewards nor penalties from the environment (see Fig. 11).

**MultiRoomLava** The agent has four actions: left, right, forward, and toggle. The doors are closed but not locked. To enter the next room, the agent has to open the closed doors with the action toggle.

**DoorKeyLava** The agent has six actions: left, right, forward, pickup, drop, and toggle. The door connecting the two rooms is locked. To enter the room with the target element, the agent should pick up the key, then go to the door, unlock it, and open it. The long action sequence makes the task difficult for exploration-based learning.

---

[2]https://minigrid.farama.org/environments/minigrid/

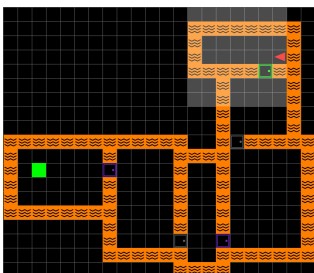 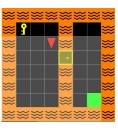 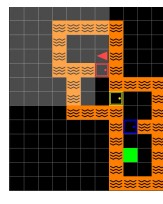 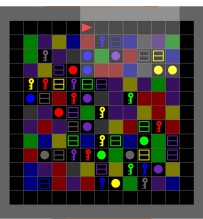

Figure 11: MiniGrid tasks used in our work. From left to right: MultiRoomLava-N6, DoorKeyLava-8x8, MultiRoomLava-N4S5, GamingEnv. Note that the size of the GamingEnv may differ when combined with different tasks.

**GamingEnv** This task is used as the game console. It contains randomly located objects, including floor, key, ball, and boxes. They are given random colors to increase their novelty. The agent can navigate onto the floor elements. The other elements are regarded as obstacles.

### D.2 HYPER-PARAMETERS AND IMPLEMENTATION

**Hyperparameter for GAGE** Since we have not changed the base algorithm implementations, we separately provide the additional hyperparameters introduced by GAGE. There is only one hyperparameter $\sigma_0$ in the continuous control experiments. The results with $\sigma_0 = 0.5, 0.75, 1.0$ are given in Sec. 4. There are two hyperparameters, $(\eta, i_0)$, in the discrete action implementation. We use $(4.5, 4.0)$, $(5.5, 1.5)$, and $(6.5, 3.0)$ for MultiRoomLava-N6, DoorKeyLava-8x8, and MultiRoomLava-N4S5 environments. They are tuned separately for each environment.

Table 2: Hyperparameters used for training agents in continuous control tasks.

| Hyperparameter | Value |
|---|---|
| **Algorithm** | |
| Value loss coefficient | 1.0 |
| Clip parameter ($\epsilon$) | 0.2 |
| Use clipped value loss | True |
| Desired KL divergence | 0.01 |
| Entropy coefficient | 0.01 |
| Discount factor ($\gamma$) | 0.99 |
| GAE parameter ($\lambda$) | 0.95 |
| Max gradient norm | 1.0 |
| Learning rate | 0.001 |
| Number of learning epochs | 5 |
| Number of mini-batches | 4 |
| Learning rate schedule | Adaptive |
| **Policy** | |
| Activation function | ELU |
| Actor hidden dimensions | [128, 128, 128] |
| Critic hidden dimensions | [128, 128, 128] |
| Initial noise standard deviation | 1.0 |
| **Runner** | |
| Number of steps per environment | 24 |
| Max iterations | 1500 |
| Empirical normalization | False |
| **RND** | |
| Intrinsic Reward coefficient | 1 |
| Extrinsic Reward coefficient | 2 |
| Intrinsic Reward Normalization | yes |

**Hyperparameter for algorithm with continuous action**  We use Proximal Policy Optimization (PPO) as the backbone algorithm for all the experiments. For the continuous control tasks, we adjust the implementation of rsl_rl v2.0.0[3]. We have not changed any hyperparameters for the implemented algorithms. They are kept the same for all agents for a fair comparison (see Table 2).

Table 3: Hyperparameters used for training each method in MiniGrid.

| Hyperparameter | MiniGrid |
| --- | --- |
| PPO $\gamma$ | 0.99 |
| PPO $\lambda_{\text{GAE}}$ | 0.95 |
| PPO rollout steps | 512 |
| PPO workers | 16 |
| PPO clip range | 0.2 |
| PPO training epochs | 4 |
| model training epochs | 4 |
| mini-batch size | 512 |
| entropy loss coef | $1 \times 10^{-2}$ |
| advantage normalization | yes |
| adv norm momentum | 0.9 |
| Adam learning rate | $3 \times 10^{-4}$ |
| Adam epsilon | $1 \times 10^{-5}$ |
| Adam beta1 | 0.9 |
| Adam beta2 | 0.999 |
| normalization for layers | Batch Norm |
| extrinsic reward coef | 1.0 |
| *DEIR* | |
| IR (intrinsic reward) coef $\beta$ | $1 \times 10^{-2}$ |
| IR normalization | yes |
| IR norm momentum | 0.9 |
| observation queue size | $1 \times 10^{5}$ |
| *Label Smoothing* | |
| Weight $\alpha_0$ for MultiRoomLava-N6 | 0.15 |
| Weight $\alpha_0$ for MultiRoomLava-N4 | 0.35 |
| Weight $\alpha_0$ for DoorKeyLava | 0.55 |
| *RND* | |
| IR coefficient $\beta$ | $3 \times 10^{-3}$ |
| IR normalization | yes |
| IR norm momentum | 0.9 |
| RND error normalization | no |
| RND error momentum | total avg |
| *ICM* | |
| IR coefficient $\beta$ | $1 \times 10^{-2}$ |
| IR normalization | yes |
| IR norm momentum | 0.9 |
| forward loss coef. | 0.2 |

**Hyperparameter for algorithm with discrete action**  For the experiments with MiniGrid, we implement the algorithm based on the code provided in DEIR[4]. We have not changed the original code except for adding our GAGE implementation. Hyperparameters are also kept unchanged to ensure a fair comparison with the baseline algorithms (see Table 3).

---

[3]https://github.com/leggedrobotics/rsl_rl
[4]https://github.com/swan-utokyo/deir

