# OpenReview forum: "Goal Achievement Guided Exploration: Mitigating Premature Convergence in Reinforcement Learning"
_ICLR.cc/2025/Conference — Submitted to ICLR 2025_

### Official Review · Reviewer_fWw4 · 2024-10-30

**Soundness:** 3
**Presentation:** 3
**Contribution:** 2
**Rating:** 5
**Confidence:** 4

**Summary:**

This paper proposes an exploration approach in reinforcement learning aimed at mitigating the premature convergence issue. The proposed Goal Achievement Guided Exploration (GAGE) measures the ratio of currently achieved cumulative rewards over the expected maximum cumulative rewards as a criterion. If the agent has not reached an expected level of performance, it is encouraged to continue exploring.

**Strengths:**

The GAGE algorithm is straightforward and easy to understand. The core idea is to set an expected "goal" and have the agent keep exploring (rather than converging) until the set goal is reached. The presentation is smooth, and the paper provides a comprehensive review of related works. The targeted issue of premature convergence is clearly stated and effectively addressed. The paper discusses both continuous and discrete action spaces and proposes appropriate solutions for each.

**Weaknesses:**

1. The "goal achievement" is defined as the ratio of achieved cumulative rewards for the current policy over the maximum or optimal cumulative rewards. Since the optimal policy is unknown, the paper proposes setting a hyperparameter as a threshold. However, this introduces two limitations:

(1) The goal-setting determines the upper bound of learning performance, or at least, heavily influence the learning process. If the goal is set too high, the algorithm may struggle to converge as the agent will always perceive its performance as insufficient. Conversely, if the goal is set too low, the agent will reach it too easily, which may still lead to premature convergence.

(2) In this case, the expected "goal" is highly task-specific, requiring prior knowledge to define an appropriate threshold for different tasks.

2. The paper identifies four main factors contributing to premature convergence (discussed in Section 2.1). However, the five continuous control tasks used in the experiments do not seem to reflect these factors well. The motivation for selecting these tasks, and how they are capable of demonstrating the effectiveness of the GAGE algorithm in addressing premature convergence, should be more clearly explained.

3. In the experiments, the five continuous control tasks only compare GAGE with the backbone PPO algorithm. I believe comparisons with some benchmarks are necessary to fully demonstrate the advantages of GAGE.

**Questions:**

1. In Section 4.1 (around Line 400), the experiments show that "When the target speed is set to 5m/s, which is below the learned optimal speed (~7m/s), the GAGE agent is still able to learn the optimal speed." Referring to Equations (2) and (4), if the learned policy achieves higher rewards than the expected target, the "goal achievement" $g(\pi) >1$, which means the lower bound $\sigma_L(\pi) = -\sigma_0 g(\pi) + \sigma_0 < 0$. Additionally, if the learned policy has already achieved the target goal of 5m/s, it would focus mainly on convergence and less on exploration, how is it able to continue optimizing to reach 7m/s?

2. What happens if the target goal is set too high? Will this result in the agent lacking confidence and failing to converge?

3. While GAGE is designed to avoid local optima, in Figure 3, we can observe that some GAGE variants still become trapped in local optima. For instance, in *Ant Acrobatics*, both GAGE-75 and GAGE-100; in *Humanoid Pole*, GAGE-100; and in *Humanoid Tightrope*, both GAGE-50 and GAGE-100 converge to relatively low episodic returns. Does this indicate that the local optima issue is not fully addressed?

I would like to increase the score if these concerns are addressed.

---

> ### Author Response · Authors · 2024-11-21
>
> We thank the reviewer for reviewing our work and providing insightful feedback. We provide further clarifications below.
>
> ## Weaknesses
>
> > "If the goal is set too high, the algorithm may struggle to converge as the agent will always perceive its performance as insufficient. Conversely, if the goal is set too low, the agent will reach it too easily, which may still lead to premature convergence."
>
> We acknowledge the reviewer's point that setting goals that are too high or too low can theoretically hinder GAGE's policy convergence due to inappropriate goals. To address this, we conducted an additional experiment (see Fig. 4(a) for the Humanoid task) as an ablation study. We tested target speeds of 0.1 m/s, 1 m/s, 20 m/s, and 100 m/s to encompass unachievable and overly simple goals. Our method demonstrated robust learning at both 1 m/s and 20 m/s. At 0.1 m/s, GAGE performed comparably to standard PPO, while at 100 m/s, GAGE outperformed standard PPO. This highlights our method's robustness. While convergence issues may still occur with extreme goals, we suggest adjusting and restarting the training with a refined goal to achieve optimal performance. This approach is more practical and efficient than tuning reward component weights to guide the exploration, which can result in a prohibitively large search space.
>
> > "the expected ''goal'' is highly task-specific, requiring prior knowledge to define an appropriate threshold for different tasks"
>
> Prior knowledge is beneficial for GAGE but not essential. In cases where no prior knowledge of an appropriate threshold for setting the goal is available, we conducted an experiment with varying target speeds, as described in our response to the previous weakness point. Moreover, in situations where access to individual reward terms is unavailable, and there is no prior knowledge of the optimal episodic returns, the episodic returns from state-of-the-art (SOTA) methods can be used as an estimate for the optimal goal. To demonstrate this, **we conducted an ablation study (see Fig. 4(b)) using 1x, 2x, and 3x (corresponding to 20, 40, and 60 episode rewards, respectively) of standard PPO episodic rewards as the goal**. The results show that GAGE can further enhance performance.
>
> > "The motivation for selecting these tasks, and how they are capable of demonstrating the effectiveness of the GAGE algorithm in addressing premature convergence, should be more clearly explained"
>
> We acknowledge the reviewer's concern that the connection between the four proposed factors (e.g., non-convexity) and the selected experimental tasks could be made clearer. We revised the motivation in the experiment setup part (Section 4.1) to connect to these factors better. Here is a further explanation of how these factors are represented in our tasks:
>
>   - Non-convexity: This is prevalent across all control tasks due to the non-convex dynamics of the robot and its interactions with the environment, resulting in a non-convex reward function landscape.
>
>   - Reward shaping: As demonstrated in our ablation study ``Improved Robustness to Reward Shaping'' in Section 4.1 (Fig. 4(c)), we modify the action penalty terms. Additionally, in the Minigrid task, the use of intrinsic rewards introduces reward shaping, which creates new local optima for the agent to navigate in learning the optimal policy.
>
>   - Multi-objectives: These are evident in the Humanoid Running task, where the reward function comprises multiple components, including speed rewards, energy penalties, and other terms.
>
>   - Approximation error: This factor is present in all tasks involving neural networks, especially in challenging environments. Here, the agent may struggle due to exploration difficulties and limitations in accurately approximating the value function, as discussed in [1].
>
> > "comparisons with some benchmarks are necessary to fully demonstrate the advantages of GAGE"
>
> We agree with the reviewer that other algorithms, such as SAC, would serve as a good baseline. Due to the time limit, we will not be able to run the experiments during the rebuttal period. But we will add this in the camera-ready version.
>
>
> [Continued in second post due to character limit]

---

> ### Author Response · Authors · 2024-11-21
>
> ## Questions
>
> > In Section 4.1 (around Line 400), the experiments show that "When the target speed is set to 5m/s, which is below the learned optimal speed (~7m/s), the GAGE agent is still able to learn the optimal speed." Referring to Equations (2) and (4), if the learned policy achieves higher rewards than the expected target, the "goal achievement" $g(\pi) >1$, which means the lower bound $\sigma_L(\pi) = -\sigma_0 g(\pi) + \sigma_0 < 0$. Additionally, if the learned policy has already achieved the target goal of 5m/s, it would focus mainly on convergence and less on exploration, how is it able to continue optimizing to reach 7m/s?
>
> It is possible that the $\sigma$ lower bound is below 0. This would make GAGE equivalent to standard PPO training, as the lower bound would no longer influence the learning process. However, due to the stochastic nature of the learning process, the $\sigma$ values would still remain greater than 0. Furthermore, compared to standard PPO, GAGE maintains a relatively larger standard deviation in the policy, thanks to its slower exploitation during the early stages, enabling continued proper exploration. For reference, **we included the plot of standard deviation for this experiment in the Appendix Fig. 6.**
>
> > "What happens if the target goal is set too high? Will this result in the agent lacking confidence and failing to converge?"
>
> This question is closely related to the first point discussed in the weakness section. As our response to that point, if the goal is set too high, the agent will maintain a high level of exploration and may not converge fully. However, it can still develop a reasonable policy by adjusting the mean of the Gaussian policy or the probability distribution of the categorical discrete policy, as the exploration level remains upper-bounded (e.g., by $\sigma_0$ and $i_0$ in continuous and discrete cases, respectively).
>
> > "While GAGE is designed to avoid local optima, in Figure 3, we can observe that some GAGE variants still become trapped in local optima. For instance, in Ant Acrobatics, both GAGE-75 and GAGE-100; in Humanoid Pole, GAGE-100; and in Humanoid Tightrope, both GAGE-50 and GAGE-100 converge to relatively low episodic returns. Does this indicate that the local optima issue is not fully addressed?"
>
> First of all, we renamed GAGE-50, GAGE-75, and GAGE-100 to GAGE-0.5, GAGE-0.75, and GAGE-1.0 for clarity as these numbers represent different values of $\sigma_0$ defined in Equation 4.
> Regarding the question, the suboptimal performance noted by the reviewer arises from two key factors. First, when $\sigma_0$ is set too high, such as 0.75 or 1.0 in Ant Acrobatics, 1.0 in Humanoid Pole, and 1.0 in Humanoid Tightrope, the agent experiences over-exploration, resulting in a slower convergence speed. This is evident in the plots, where the episode return curves for these settings continue to increase gradually, even toward the end of training. Second, when $\sigma_0$ is set too low, such as 0.5 in Humanoid Tightrope, the agent quickly over-exploits, converging to suboptimal local policies similar to those observed with standard PPO. While our method can partially mitigate issues related to local optima, we acknowledge that fully addressing these challenges will require further investigation and development in future work.
>
> We hope this helps in clarifying any questions the reviewer might have. We are happy to provide further clarification to any other pending concerns and suggestions and to further improve our work.
>
> [1] Nikishin, Evgenii *et al.* ``The Primacy Bias in Deep Reinforcement Learning.'' International Conference on Machine Learning (2022).

---

> > ### Comment · Reviewer_fWw4 · 2024-11-25
> >
> > Many thanks for the authors' detailed reply.
> >
> > I have a follow-up question:
> >
> > For the claim:
> >
> > > It is possible that the $\delta$ lower bound is below 0. This would make GAGE equivalent to standard PPO training, as the lower bound would no longer influence the learning process.
> >
> > If the "goal achievement" $g(\pi) > 1$, then the $\delta < 0$, then GAGE will be equivalent to the PPO algorithm, in this case, in Figure 4(a), the new ablation study, for target speed = 0.1 or 1, which is quite easy to achieve, the GAGE will be PPO, but can you explain why it still outperforms the PPO over 1.6~2 times of the performance?
> >
> > Besides, I think some of the concerns in my initial reviews are still not addressed, so I want to maintain my score:
> >
> > 1. the paper's kernel idea is "setting a target, if not achieved, then explore longer time using random actions", my main concern is: the exploration method itself is not improved. Simply making longer exploration time doesn't mean making better/broader exploration. In other words, GAGE forces the agent to extend the exploration time, but it doesn't ensure the range of exploration is wider. In contrast, approaches like curiosity-driven [1] and novelty-based [2,3] exploration improve the range of exploration. (That's why I'm looking forward to some comparison with exploration baselines). From another perspective, could PPO achieve similar effects by simply setting a longer burn-in period?
> >
> > 2. The performance of GAGE seems very sensitive to the target setting or the maximum reward estimating, as shown in Figure 3. The authors explained this from two points: (1) the target is set too high, leading to over-exploration, and some curves are still increasing; (2) the target is set too low, leading to over-explots. In this case, the performance of GAGE highly depends on the target setting, and in some environments, without prior knowledge, we don't know if a target is set too high or too low.
> >
> > 3. The paper didn't compare with any baselines, except their backbone PPO, which didn't effectively show GAGE's outperformance. GAGE is a work studying "exploration", I believe at least some exploration algorithms should be compared, for example, curiosity-driven exploration[1], novelty-rewarded exploration, e.g., the famous random network distillation [2,3], reward-shaping based [4,5], etc. The authors replied that:
> >
> > > We agree with the reviewer that other algorithms, such as SAC, would serve as a good baseline. Due to the time limit, we will not be able to run the experiments during the rebuttal period. But we will add this in the camera-ready version.
> >
> > The plan of these experiments in the camera-ready version cannot be guaranteed, and more importantly, the results of comparisons with these baselines are unknown.
> >
> > [1] Pathak, Deepak, et al. "Curiosity-driven exploration by self-supervised prediction." International conference on machine learning. PMLR, 2017.
> >
> > [2] Burda, Yuri, et al. "Exploration by random network distillation." arXiv preprint arXiv:1810.12894 (2018).
> >
> > [3] Yang, Kai, et al. "Exploration and anti-exploration with distributional random network distillation." arXiv preprint arXiv:2401.09750 (2024).
> >
> > [4] Devidze, Rati, Parameswaran Kamalaruban, and Adish Singla. "Exploration-guided reward shaping for reinforcement learning under sparse rewards." Advances in Neural Information Processing Systems 35 (2022): 5829-5842.
> >
> > [5] Sorg, Jonathan, Richard L. Lewis, and Satinder Singh. "Reward design via online gradient ascent." Advances in Neural Information Processing Systems 23 (2010).

---

> ### Author Response · Authors · 2024-11-27
>
> We appreciate the reviewer’s follow-up questions. Below, we aim to address the reviewer’s concerns and provide answers to the raised questions.
>
> > "in Figure 4(a), the new ablation study, for target speed = 0.1 or 1, ..., why it still outperforms the PPO over 1.6~2 times of the performance?"
>
> This phenomenon highlights the critical importance of exploration during the initial stages of training. As shown in Fig. 6(b), at the very beginning, none of the agents had learned to move forward, i.e. $g(\pi)\approx 0$. As a result, even with small target speeds of 0.1 or 1, the GAGE agent maintained higher standard deviation values compared to standard PPO. During this period, the PPO agent drastically reduced the standard deviation by over-exploiting auxiliary rewards. Since the optimal locomotion gait for the humanoid robot remains consistent across different speeds, it is crucial for the agent to avoid becoming trapped in a suboptimal policy (gait) early in training. Once the agent learns an effective low-speed gait, gradually increasing locomotion speed with a similar gait requires less exploration. This observation also suggests that the optimal relationship between goal achievement and exploration may not be linear and could vary across tasks. As noted in the paper, *investigating non-linear relationships between goal achievement and exploration metrics, such as the standard deviation of Gaussian distributions in continuous action spaces, could further enhance the method’s adaptability to diverse RL problems.* Nevertheless, the current linear relationship has already demonstrated strong performance by introducing adaptive exploration.
>
> > "setting a target, if not achieved, then explore longer time using random actions"
>
> The exploration is not entirely random. Instead, the agent explores around the learned mean actions with a lower-bounded variance when the goal has not yet been achieved. Importantly, the mean ($\mu(s)$) of the action distribution remains unconstrained, allowing it to continue learning meaningful values even with the lower-bounded variance. This approach ensures a balanced trade-off between exploration and exploitation, preventing the agent from either exploring completely randomly or converging prematurely.
>
> > "GAGE forces the agent to extend the exploration time, but it doesn't ensure the range of exploration is wider."
>
> GAGE is not designed to directly widen the range of exploration or replace existing exploration approaches but to complement them by adaptively lower-bounding the exploration range. In both continuous and discrete tasks, GAGE introduces an adaptive **lower bound** for the exploration level, making it compatible with methods like entropy maximization and intrinsic rewards. Its primary contribution is addressing the premature convergence of other exploration methods by incorporating prior knowledge of the agent's performance. For instance, as highlighted in the MiniGrid experiments, *our method builds on DEIR*. Here, DEIR’s intrinsic rewards remain responsible for encouraging the exploration of novel states. Meanwhile, GAGE ensures an adaptive lower bound for the exploration level, helping to prevent the DEIR agent from being distracted by phenomena such as Noisy-TV or Game Console effects. By focusing on complementing existing methods rather than replacing them, GAGE enhances their robustness and mitigates the risks of premature convergence.
>
> [Continued in the second post due to character limit]

---

> > ### Author Response · Authors · 2024-11-27
> >
> > > "That's why I'm looking forward to some comparison with exploration baselines."
> >
> > We apologize for misunderstanding the reviewer's intent in the first review. We initially believed the reviewer was requesting results with different RL algorithms such as SAC, DDPG, etc. Upon clarification, since the reviewer is requesting comparisons with exploration algorithms, **we have added a baseline with RND [1] in all continuous tasks**, as suggested by the reviewer. We followed the original hyperparameter settings of RND with a 2:1 ratio of extrinsic-to-intrinsic reward weights [1,2].
> > As shown in Figure 3, RND failed to solve any of the tasks. To further investigate the effect of intrinsic rewards, **we have also conducted a hyperparameter tuning for RND on the Dog Balance Beam task as an example**, included in the appendix. In Figure 7 of the appendix, we demonstrate that all RND agents fail to solve the task. Agents with larger ratios of extrinsic-to-intrinsic weights exhibit learning behaviors similar to standard PPO. As this ratio decreases, agents focus more on exploring novel states, as indicated by the larger standard deviations during training. However, this increased exploration does not help agents solve the task. Instead, novelty-based exploration leads to a reduction in extrinsic rewards.
> > This phenomenon highlights the distinct focus of our work compared to curiosity- or novelty-based exploration methods. As noted in the introduction, *there are two prominent challenges in exploration: sparse reward functions and local optima.* Our work focuses on addressing premature convergence, an issue that is equally important but has been largely overlooked until now. In contrast, curiosity-based methods primarily tackle sparse rewards.
> > The difference in focus is also reflected in the existing benchmarks for exploration algorithms. Most environments are designed with sparse rewards and moderate local optima, which can be effectively addressed using novelty-based exploration. For example, environments like Fetch [3], MiniGrid [4], AntMaze, and Adroit manipulation tasks [5] are "safe," with sparse termination states or penalties distributed across the state space. Agents can easily avoid termination and penalty states while exploring for rewards. In such environments, exploring unseen states is a highly effective strategy.
> > However, novelty-based methods struggle in scenarios with more severe and deeper local optima. For instance, Noisy-TV has been recognized as a major issue for novelty-based methods, even though it only involves local optima introduced by environment stochasticity. The challenges posed by more severe local optima have not yet been fully explored.
> > In this work, we aim to push the boundaries of RL exploration research into environments with more challenging local optima issues. The IsaacLab tasks reflect real-world robot control scenarios where optimal behaviors occupy only a small portion of the state space, while most of the state space leads to penalties such as falling down or wasting energy. This dominant penalizing space creates challenging local optima. In such environments, novelty-based exploration often results in sampling mostly failed trajectories and becoming trapped in local optima.
> > A similar phenomenon is observed in the MiniGrid experiments, where popular novelty-based methods fail to solve tasks with more challenging local optima. We encourage the reviewer to review our experiment results for discrete action spaces, where we compare our method with several popular exploration techniques, including RND, ICM, and DEIR.
> >
> > > "From another perspective, could PPO achieve similar effects by simply setting a longer burn-in period?"
> >
> > This statement is reasonable. However, in practice, determining the schedule for such a burn-in period is highly task-specific and may require extensive tuning. This is evident in the policy standard deviation plots in Figure 3, where the plateau stages of the GAGE agents can be viewed as the "burn-in" periods suggested by the reviewer. The length of these periods varies significantly across different tasks and even across different seeds for the same task, making it challenging to define a general and effective burn-in period. In contrast, GAGE provides an adaptive "burn-in" period that adjusts dynamically based on the agent's performance. Notably, GAGE achieves this adaptability with the same settings (e.g., GAGE-0.75) applied consistently across all tasks, reducing the need for extensive task-specific tuning.
> >
> > [Continued in the third post due to character limit]

---

> > > ### Author Response · Authors · 2024-11-27
> > >
> > > > "GAGE seems very sensitive to the target setting or the maximum reward estimating"
> > >
> > > Contrary to the reviewer's concern, we emphasize the robustness of our method against varying target settings. As noted by the reviewer, our method significantly outperforms standard PPO across a wide range of target speeds, from 0.1 to 100 m/s, as demonstrated in Figure 4(a) for the humanoid locomotion task. These results illustrate that GAGE maintains strong performance even under diverse target settings.
> > >
> > > > "The paper didn't compare with any baselines, except their backbone PPO" and "The plan of these experiments in the camera-ready version cannot be guaranteed."
> > >
> > > As mentioned above, **we have added a new baseline Random Network Distillation (RND) in our continuous control tasks**, with the results presented in Fig. 3. **We have also conducted a hyperparameter tuning for RND to investigate the effect of intrinsic rewards**. Additionally, we encourage the reviewer to examine the experimental results for discrete action spaces, where we compared our method against several popular exploration techniques, including curiosity-driven exploration [6], Random Network Distillation [1], and Discrimination-Model-Based Episodic Intrinsic Rewards (DEIR) [7]. As demonstrated in the discrete task results, novelty-based methods can introduce new local optima due to a mismatch between the goal reward and intrinsic reward. These findings further support the importance of our approach in addressing such challenges.
> > >
> > > To conclude, we would like to gently emphasize that the primary focus of this work is **not on addressing sparse rewards**, which has been the central aim of most existing novelty-based exploration methods. Instead, our goal is to **tackle the issue of premature convergence, as detailed in the first paragraph of Section 1, an equally important yet largely overlooked challenge in reinforcement learning.** We hope this distinction clarifies our contribution and the unique perspective of our approach.
> > >
> > > [1] Burda, Yuri, et al. "Exploration by random network distillation." ICLR, 2019
> > >
> > > [2] Yang, Kai, et al. "Exploration and anti-exploration with distributional random network distillation." arXiv preprint arXiv:2401.09750, 2024
> > >
> > > [3] Matthias Plappert, et al. "Multi-Goal Reinforcement Learning: Challenging Robotics Environments and Request for Research." https://arxiv.org/pdf/1802.09464, 2018
> > >
> > > [4] Maxime Chevalier-Boisvert, et al. "Minigrid \& Miniworld: Modular \& Customizable Reinforcement Learning Environments for Goal-Oriented Tasks." NeurIPS, 2023
> > >
> > > [5] Justin Fu, et al. "D4rl: Datasets for deep data-driven reinforcement learning." https://arxiv.org/pdf/2004.07219, 2021
> > >
> > > [6] Pathak, Deepak, et al. "Curiosity-driven exploration by self-supervised prediction." International conference on machine learning. PMLR, 2017
> > >
> > > [7] Shanchuan Wan et al., "DEIR: Efficient and Robust Exploration through Discriminative-Model-Based Episodic Intrinsic Rewards." IJCAI, 2023

---

> > > > ### Comment · Reviewer_fWw4 · 2024-12-01
> > > >
> > > > Thank you to the authors for their detailed responses and answering my questions. Many of my concerns have been explained.
> > > >
> > > > However, my main concern remains the target value $\sigma_0$, the most important hyperparameter in the paper. First, setting an appropriate target value heavily influences the convergence performance, convergence speed, and the number of training steps. This value appears to be task-specific, meaning that prior knowledge of the achievable returns is necessary for better parameter tuning. While the authors suggested one possible approach:
> > > >
> > > > > The episodic returns from state-of-the-art (SOTA) methods can be used as an estimate for the optimal goal. To demonstrate this, we conducted an ablation study (see Fig. 4(b)) using 1x, 2x, and 3x (corresponding to 20, 40, and 60 episode rewards, respectively) of standard PPO episodic rewards as the goal.
> > > >
> > > > This still requires prior knowledge of how other methods perform in the same environment, which is not typically required by other algorithms.
> > > >
> > > > More importantly, in the experiments provided by the authors, I continue to struggle to understand **why GAGE consistently outperforms both PPO and RND across various target values, even when extreme values (e.g., 0.1, 100) are set**. This makes it challenging to comprehend the exact role of the target value, as it appears that its value does not matter, yet GAGE consistently achieves at least 1.5x better performance than the baselines. This behavior seems counterintuitive and may lead to the impression that the superior performance is not actually driven by the target value. For example, introducing a fixed lower bound to encourage higher variance in the stochastic policy might achieve similar effects. I believe this point warrants further investigation or theoretical justification.
> > > >
> > > > Given this, I would like to maintain my current score. Thanks again for the authors' response.

---

> ### Author Response · Authors · 2024-12-02
>
> We appreciate the reviewer’s acceptance of most of our explanations. Below, we address the remaining questions raised by the reviewer:
>
> > However, my main concern remains the target value $\sigma_0$ ...
>
> We believe there may be a misunderstanding. As stated in the paper, *$\sigma_0$ is a hyperparameter controlling the minimum allowed $\sigma$ value when the goal achievement is zero.* However, the "target value" referred to by the reviewer may correspond to the *optimal value of the goal reward $r_{\max, t}$* instead. The hyperparameter $\sigma_0$ does not require extensive task-specific tuning. Empirically, either values of $0.5$ or $0.75$ works well across all benchmark tasks.
>
> >  setting an appropriate target value heavily influences the convergence performance ... This still requires prior knowledge of how other methods perform in the same environment, which is not typically required by other algorithms.
>
> We do not believe this is a critical flaw. Many general algorithms, such as RND, require task-specific prior knowledge. For example, practitioners must balance extrinsic and intrinsic reward scales, as shown in Fig. 7, where different ratios significantly affect performance. GAGE introduces fewer hyperparameters than RND. We provide default values for $\sigma_0$ (e.g., $0.75$), and the optimal goal reward can often be derived from the reward function or task-specific knowledge, such as the desired speed of a robot. For scenarios without prior knowledge, we provide a task-agnostic method using PPO to determine the optimal goal. This simplifies hyperparameter tuning.
>
> While the idea of a general algorithm with fixed hyperparameters is appealing, reinforcement learning algorithms are well-known to be sensitive to hyperparameters [1]. Incorporating task-specific prior knowledge has consistently improved training efficiency across diverse tasks [2,3,4,5,6,7].
>
> > More importantly, in the experiments provided by the authors ... yet GAGE consistently achieves at least 1.5x better performance than the baselines.
>
> This phenomenon underscores the critical role of exploration in the well-known humanoid locomotion task. GAGE enhances the exploration level whenever the target speed is greater than zero, as shown in Fig. 6(b), enabling it to outperform PPO across all settings. For extremely small target speed values (e.g., 0.1), our method maintains higher standard deviations than standard PPO, as illustrated in Fig. 6(b). This increased exploration is especially crucial during the early stages of training, when the robot learns basic movements such as standing but not yet walking. GAGE prevents the robot from being trapped in suboptimal policies that over-exploit rewards like $r_\text{alive}$. For example, if the robot initially learns to balance by standing on its heels while bending backward, it would struggle to adapt to leaning forward and running if $\sigma$ were already too small. For extremely large target speed values (e.g., 100), while GAGE keeps $\sigma$ close to $\sigma_0$, the action mean $\mu(s)$ remains unconstrained. This allows policies with well-learned $\mu(s)$ to perform effectively, even with the action noise introduced by higher $\sigma$. This is similar to domain randomization techniques commonly used in Sim2Real applications. However, for more challenging tasks, such as those we proposed, the range of target speeds yielding near-optimal performance becomes narrower. If the reviewer would like to see corresponding experimental results, we will include them in the camera-ready version.
>
> [Continued in the next post due to character limit]

---

> ### Author Response · Authors · 2024-12-02
>
> >  This behavior seems counterintuitive and may lead to the impression that the superior performance is not actually driven by the target value. For example, introducing a fixed lower bound to encourage higher variance in the stochastic policy might achieve similar effects. I believe this point warrants further investigation or theoretical justification.
>
> This is not supported by the results of our additional experiment "$\sigma$ schedule" in the appendix, as shown in Fig. 6(a). Fixed lower bounds are ineffective because a proper $\sigma$ schedule requires higher values at the beginning to enable sufficient exploration and lower values toward the end to allow convergence to optimal policies. Predefined entropy schedules, while theoretically feasible, demand extensive tuning of both the entropy values and the training duration, which can be computationally expensive.
> In contrast, our method introduces an adaptive schedule that dynamically adjusts $\sigma$ lower bound, significantly reducing the tuning workload while ensuring robust performance across tasks.
>
> We would thank the reviewer's feedback. Hope these answers could address the rest concerns.
>
> [1] Henderson, Peter, et al. "Deep reinforcement learning that matters", AAAI, 2018.
>
> [2] Peng, Xue Bin, et al. "DeepMimic: example-guided deep reinforcement learning of physics-based character skills", ACM Transactions On Graphics, 2018
>
> [3] Pertsch, Karl, et al. "Accelerating reinforcement learning with learned skill priors", CoRL, 2020
>
> [4] Peng, Xue Bin, et al. "AMP: Adversarial Motion Priors for Stylized Physics-Based Character Control", ACM Transactions On Graphics, 2021
>
> [5] Singh, Avi, et al. "Parrot: Data-driven behavioral priors for reinforcement learning", ICLR, 2021
>
> [6] Wang, Dian, et al. "$\mathrm {SO}(2) $-Equivariant Reinforcement Learning", ICLR, 2022
>
> [7] Huang, Haojie, et al. "Fourier Transporter: Bi-Equivariant Robotic Manipulation in 3D", ICLR, 2024

---

### Official Review · Reviewer_QVip · 2024-11-04

**Soundness:** 3
**Presentation:** 2
**Contribution:** 3
**Rating:** 6
**Confidence:** 3

**Summary:**

This paper proposes to use goal achievement as a learning progress measure to schedule the noise for exploration, where goal achievement is defined as the ratio of the current policy's expected return to the optimal policy's expected return. The results showed that goal achievement improves PPO's performance in robotic tasks with intensive reward shaping and hard-exploration tasks in MiniGrid.

**Strengths:**

This method is very easy to implement and seems to improve the performance greatly. The authors claim that current methods suffer from premature convergence. Thus, they propose to tune the noise of exploration adaptively using a goal achievement rate, with the assumption that the maximum reward is known.

**Weaknesses:**

- Lack of theoretical discussion. This is fine since I understand this paper's contribution is a practical algorithm. Still, it would be great to see why adjusting the noise level with goal achievement leads to improvement.
- Writing is a bit verbose. Section 2 is mostly about previous works. The proposed method doesn't come until page 4, which is too long in my opinion.

**Questions:**

- Figure 3's legends are too small to read.
- Where are the differences between GAGE-50 and GAGE-100?

---

> ### Author Response · Authors · 2024-11-21
>
> We thank the reviewer for reviewing our work and providing insightful feedback. The reviewer raises several important questions that we address in order:
>
> ## Weaknesses
> > "Lack of theoretical discussion"
>
> We agree with the reviewer that our current version lacks a theoretical foundation. Regarding reward maximization, our method does not alter the objective or reward function, so it retains the same performance guarantees as standard PPO. As for the convergence rate, our current work is mainly empirical, and we plan to address this aspect in future research.
>
> > "Writing is a bit verbose. Section 2 is mostly about previous works"
>
> Section 2 includes our problem statement and related work. Since we consider premature convergence to be the main research problem we aim to address, we dedicated a specific part to explain it. We agree with the reviewer that the current structure lacks clarity. To improve this, we renamed the second part to "Related Work".
>
> ## Questions
> > "Figure 3's legends are too small to read"
>
> We adjusted the legend font size of Fig. 3 for better clarity.
>
> > "The differences between GAGE-50 and GAGE-100"
>
> We renamed GAGE-50, GAGE-75, and GAGE-100 to GAGE-0.5, GAGE-0.75, and GAGE-1.0 for clarity as these numbers represent different values of $\sigma_0$ defined in Equation 4. We also added an explanation in Sec. 4.1.
>
> We hope this is helpful to clarify the reviewer's concerns. We’re happy to address any remaining questions from the reviewer and to make further improvements to our work.

---

> > ### Comment · Reviewer_QVip · 2024-11-24
> >
> > Thanks for the response. That addressed my questions. I will keep the rating.

---

### Official Review · Reviewer_DiJ6 · 2024-11-04

**Soundness:** 4
**Presentation:** 4
**Contribution:** 3
**Rating:** 8
**Confidence:** 4

**Summary:**

This paper presents Goal Achievement Guided Exploration (GAGE), an algorithm to prevent premature convergence and encourage exploration in deep RL. The paper describes the major causes of premature convergence in RL and describes an algorithm to address specific types of issues, which is then evaluated in several different task domains.

**Strengths:**

This paper is well written, and does an excellent job of contextualizing and motivating work on premature convergence, and intuitively develops and explains the GAGE algorithm, which is simple yet not trivial. This issue is an important one that is common in practice, but has received little attention from prior work, and thus a worthwhile topic of study. The GAGE algorithm requires several well-stated priors to work, most notably a human estimate of what the maximum achievable reward is for each reward term, but in my option this is a reasonable prior for many RL domains and not overly restrictive. The experimental validation of the algorithm is reasonably thorough.

**Weaknesses:**

I didn't have any major issues with this paper, though there's a few issues I've noted in the questions section which could be improved.

The biggest concern I have is that the benefits of doing any form of action smoothing to prevent premature convergence versus the specific algorithm of GAGE are not clear- it could be the case that a simpler baseline would be just as good (though I suspect this is not the case). However, this paper does not claim to be definitive regarding premature convergence prevention or action smoothing algorithms (and it doesn't need to be to provide a meaningful contribution), so I don't find this to be a critical flaw.

While not the final word on the problem (if such a thing is even possible), this work seems like a worthwhile step forward, with real implications for deep RL in practical and scientific use. I could see GAGE or a similar algorithm plugging in nicely as a standard tool to improve performance and stability alongside other methods. As such, I am inclined to recommend acceptance- this is good work.

**Questions:**

Some minor issues and questions:

-What do the upper and lower brackets in equation 7 denote? I don't see this explained in the text and it is unusual notation in my experience of the field.

-The temperature computation for smoothing action probabilities is somewhat complex compared to simpler alternatives mentioned (e.g. mixing with uniform). I don't see any ablations testing whether this more sophisticated smoothing is better than the naive baseline, however, which be useful to see.

-The lines in figure 3 are a bit too small to comfortably read, please make them bigger (plot size is fine, the lines are too narrow).

-The captioning and plot spacing in figure 4 is a little confusing, the right two plots should be closer together to show that they are a pair, unlike the left plot.

-Section 4.2's writing takes a sudden nosedive- there's a number of instances of odd phrasing and wrong grammar here in what is otherwise an excellently written paper. This could use a pass to revise.

-For figure 5, what is the baseline performance for each method on the non-game-containing version of this task? I assume all algorithms can learn the task successfully? It would be good to make this clear if it is so as it strengthens the point being made.

-I would have liked to see more aggressive stress tests on the reward upper bound estimate where performance is lost as a result of a bad estimate. What happens if V_star in figure 4a is set to 1? What if it is set to 99? I imagine these won't perform well, but it would be useful to know what happens when things break down since sometimes human estimates of the maximum possible reward will be quite wrong.

---

> ### Author Response · Authors · 2024-11-21
>
> We are happy to hear that the reviewer found our work thorough and our method effective. We thank the reviewer for the supportive review. The reviewer raises several important questions that we address in the following:
>
> ## Questions
>
> > "What do the upper and lower brackets in equation 7 denote?"
>
> The upper and lower brackets denote the ceiling and floor function for a real number. We agree with the reviewer that this should be clearly defined in the paper and added the explanation next to Equation 7.
>
> > "compared to simpler alternatives mentioned (e.g., mixing with uniform)"
>
> **We introduced an additional baseline using Label Smoothing (by mixing with a uniform distribution) in the MiniGrid experiments**. The results indicate that Label Smoothing (LS) fails to solve any of the tasks. Although LS maintains high entropy when the agent has not yet achieved external rewards, it leads to less effective exploration compared to GAGE agents. This inefficiency arises because the uniform distribution keeps the probabilities of undesired actions, such as those leading to termination in Lava cells, relatively high. As illustrated in the episode length plots in Fig. 7, these findings support our hypothesis.
>
> > "The lines in Figure 3 are a bit too small"
>
> We adjusted the line width accordingly.
>
> > "The captioning and plot spacing in Figure 4 is a little confusing"
>
> We adjusted the spacing between the two plots of Fig. 4(c).
>
> >"Section 4.2's writing takes a sudden nosedive":
>
> **We revised Sec. 4.2 to improve the writing quality.**
>
> > "what is the baseline performance for each method on the non-game-containing version of this task"
>
> The baseline performance on the original non-game-containing environments can be found in the work of DEIR[1]. All of the baseline algorithms can solve MultiRoom-N4S5 and DoorKey-8x8, while only ICM can not solve MultiRoom-N6.
>
> > "more aggressive stress tests on the reward upper bound estimate"
>
> We added new results by varying the target speed to $V_*=0.1,1,9,20,100$ m/s as a stress test. Our findings demonstrate that GAGE learns the optimal speed when the target speed is set between 1 m/s and 9 m/s. Even when the target speed is set to extreme values, such as 0.1 m/s or 100 m/s, GAGE outperforms standard PPO. For more details, please refer to the updated Fig. 4(a).
>
> We hope these responses address the reviewer’s concerns. Please feel free to reach out if you have any further questions.
>
> [1] Shanchuan Wan *et al.*, DEIR: Efficient and Robust Exploration through Discriminative-Model-Based Episodic Intrinsic Rewards.

---

> > ### Comment · Reviewer_DiJ6 · 2024-11-22
> > **Response to rebuttal**
> >
> > Thanks for the response! These additions seem like they improve the paper nicely, I appreciate all your hard work! I don't think I have any additional questions at this time.
> >
> > I am happy to maintain my score, I continue to think this is a good paper.

---

### Official Review · Reviewer_dGGt · 2024-11-04

**Soundness:** 1
**Presentation:** 2
**Contribution:** 3
**Rating:** 3
**Confidence:** 4

**Summary:**

The paper proposes an approach called Goal Achievement Guided Exploration (GAGE) to address premature convergence in reinforcement learning algorithms. Instead of using intrinsic rewards for exploration, the proposed approach maintains an estimate for the optimal performance level, comparing this level to the current performance for controlling between exploration and exploitation.

The main claim of the paper is that the proposed approach enhances exploration in reinforcement learning.

**Strengths:**

The proposed approach aims to balance between exploration and exploitation in reinforcement learning. The approach is interesting in that it assumes that each reward term is equally important and exploration magnitude is kept as high as how far each reward term is from an assumed optimal solution.

In more detail, the approach uses a goal achievement term that is the minimum of goal achievement terms of each reward part. Each of these individual goal achievement terms is computed using Monte Carlo estimates of recent samples divided by a heuristic estimate of the optimal value, or, total maximum reward. An implicit assumption is that an agent should be able to succeed in all parts of the reward function sum.

Discussion of the "Game Console" problem in exploration is valuable.

**Weaknesses:**

The approach is based on assuming explicit knowledge of the reward function and  the individual parts (terms) that as a sum define the reward function. This needs to be discussed and motivated in detail. Most of the exploration approaches in reinforcement learning do not need explicit knowledge of the reward function.

The approach makes strong assumptions about the task. I assume the approach only works if these assumptions are satisfied and can easily lead to slow convergence. The approach controls exploration according to the reward term that is furthest away from being satisfied. This means, for example, that if there is a single reward term that is very hard to get close to optimal, large amounts of exploration is used although the total reward would be already high. Moreover, the approach can lead to excessive exploration noise that may hinder improving reward terms which require small amount of noise.

Evidence for the main claim of the paper that the proposed approach enhances exploration is needed. That the algorithmic design and computations used in the approach improve on state-of-the-art need significantly stronger theoretical or/and empirical evidence.

Fig. 2 and the main text aim to motivate the proposed approach by saying that exploration methods typically somehow change the order of probabilities. This is not true. For example, target entropy [Haarnoja et al., 2018] is commonly used and does not change the order of the action probabilities.

The action smoothing procedure in Section 3.2 for discrete actions includes several computations for which there is some discussion of the motivation but no theoretical or empirical evidence. There should be a much more convincing discussion on why each of the steps 1. to 4. in Section 3.2 is used to compute the adaptive temperature of the softmax distribution.


Experiments:

Methods:
One of the main motivations for the proposed approach in the paper is that intrinsic motivation based approaches may converge to local optima. However, there are methods designed specifically to address this problem. For example, [Chen et al., 2022], explicitly optimizes the original optimization objective while taking advantage of intrinsic motivation. These kind of methods need to be added as baselines.

Typical exploration methods need to be added as baselines. This includes pre-defined entropy schedules: linearly descreasing entropy, constant entropy, constant + linearly decreasing etc.

Benchmarks:
In the continuous action setting, the proposed new benchmarks are valuable. However, to provide readers sufficient information also well known benchmarks should be used where existing baseline results are available. Examples of continuous action benchmarks which require exploration such as AntMaze etc. can be found for example in the hierarchical reinforcement learning literature (see [Nachum et al., 2018] and follow the citations to the newest work with the largest environments).

The "Game Console" problem in exploration is valuable and interesting but what is the relationship of the proposed approach compared to other methods that do not use intrinsic rewards? In "Game Console" type of problems, mostly intrinsic rewards cause problems?


Details:

Please explain "More severely, for discrete actions, the entropy loss can not maintain the distribution shape, i.e., the order of actions’ probabilities of the learned policy." in more detail.

Regarding control of policy variance in Equation 4, it seems that identical variances for all action dimensions is assumed?

In Fig. 2, please define what entropy maximization means. For a discrete distribution, maximum entropy results in a uniform distribution which differs from Fig. 2b.

The presentation is overall OK but there are typos such as  "probablities" that should be fixed.


Chen, E., Hong, Z. W., Pajarinen, J., & Agrawal, P. (2022). Redeeming intrinsic rewards via constrained optimization. Advances in Neural Information Processing Systems, 35, 4996-5008.

Nachum, O., Gu, S. S., Lee, H., & Levine, S. (2018). Data-efficient hierarchical reinforcement learning. Advances in neural information processing systems, 31.

Haarnoja, T., Zhou, A., Hartikainen, K., Tucker, G., Ha, S., Tan, J., Kumar, V., Zhu, H., Gupta, A., Abbeel, P. and Levine, S., 2018. Soft actor-critic algorithms and applications. arXiv preprint arXiv:1812.05905.

**Questions:**

I recommend rejecting the paper. The authors can improve the paper by improving the motivation for the approach, discussing in more detail in which situations the approach works and does not work, providing proper experimental baselines and benchmarks.

---

> ### Author Response · Authors · 2024-11-20
>
> We appreciate the reviewer’s thorough and detailed feedback. We are sorry that there seem to be a few misunderstandings regarding our work. Below, we try to clarify the reviewer’s concerns and answer the questions:
>
>
> ## Summary
>
> > "Instead of using intrinsic rewards for exploration"
>
> We believe this is a misunderstanding. Our method is not designed to replace intrinsic reward exploration, nor is it solely focused on intrinsic rewards. We also discussed other exploration techniques and factors contributing to premature convergence. Our method is developed as a general tool that can be integrated with every exploration approach to effectively address premature convergence.
>
> ## Strengths
>
> > "it assumes that each reward term is equally important"
>
> > "An implicit assumption is that an agent should be able to succeed in all parts of the reward function sum"
>
> We hope that the following responses can resolve these potential misunderstandings. Our work does not make this assumption. Please note that we specifically select the most important reward term (goal reward) to calculate the agent’s performance (goal achievement) while excluding auxiliary reward terms that provide dense exploration information. This is demonstrated in Table 1 of the appendix, in which we highlight the chosen goal reward for each task. For example, in the ``Humanoid Dribbling Task'', the auxiliary reward for staying close to the football is used only to guide the robot toward interacting with the ball. It is acceptable if this reward is not maximized, which is why we do not calculate goal achievement for such auxiliary terms.
>
>
> ## Weaknesses
>
> > "Most of the exploration approaches in reinforcement learning do not need explicit knowledge of the reward function"
>
> This statement aligns with our observation that most exploration approaches do not rely on prior knowledge of the reward function. However, we believe our work serves as an important stepping stone to advance existing algorithms further. We are convinced that reinforcement learning can greatly benefit from leveraging this idea to address its well-known instability. Additionally, since reward shaping is widely regarded as an important technique in reinforcement learning, prior knowledge of reward function composition is often available in many tasks. Moreover, goal achievement can also be computed based on the total reward function rather than individual reward terms. Even when the theoretical maximum of the total reward is unknown, the learning process can still benefit from using estimated optimal reward values, as demonstrated in the **Unknown Optimal Goal** experiments. **We also added additional experiments in Section 4.1 in which we compute goal achievement using the reward function sum** on the ``Humanoid Locomotion Task''.
>
>
> > "I assume the approach only works if these assumptions are satisfied and can easily lead to slow convergence"
>
> We kindly ask the reviewer to specify which assumptions are deemed unrealistic or need improvement. Regarding convergence speed, as shown in Fig. 4(a) in the first submission and Fig. 4(b) in the revised version, our method achieves the same convergence speed as the baseline algorithm in the popular humanoid locomotion task while delivering significantly higher final performance. If the concern is related to the highly challenging tasks presented in Fig. 3, we acknowledge that the baseline algorithm often converges faster initially. However, this is because the baseline over-exploits early in the learning process, leading to suboptimal solutions. This issue is precisely what our method is designed to address. We kindly ask the reviewer to reconsider our contribution and the balance between convergence speed and final performance. For example, in the ``Humanoid Pole Task'', the baseline method fails to converge by the end of training. Its initial over-exploitation results in a much slower subsequent learning process than our approach.
>
>
> > "if there is a single reward term that is very hard to get close to optimal, large amounts of exploration is used although the total reward would be already high"
>
> > "excessive exploration noise that may hinder improving reward terms which require small amount of noise":
>
> We appreciate the reviewer's insight into the challenge of balancing exploration noise, particularly for tasks where different reward terms may benefit from varying noise levels. In our work, we adopted a minimalistic approach to hyperparameter design by using a single, identical $\sigma_0$ and taking the minimum of all goal achievements across different reward terms to ensure simplicity and consistency. This design choice provides flexibility for practitioners to adapt as needed. We acknowledge that developing adaptive or task-specific noise strategies is a promising direction and plan to explore this further in future work.
>
> [Continued in second post due to character limit]

---

> ### Author Response · Authors · 2024-11-20
>
> > "algorithmic design and computations used in the approach improve on state-of-the-art need significantly stronger theoretical or/and empirical evidence"
>
> We agree that our work introduces an empirical method without formal theoretical proof. To address the need for stronger empirical evidence, **we added new experimental results, including new baselines, ablation study, and hyperparameter tuning**. We hope that this makes the reviewer more content, although we will definitely acknowledge suggestions for further necessary experiments.
>
> > "target entropy [Haarnoja et al., 2018] is commonly used and does not change the order of the action probabilities"
>
> According to our understanding, target entropy cannot maintain the learned order of action probabilities. The target entropy mechanism introduces an automatic adjustment for the coefficient $\alpha$ in the maximum entropy objective: $\sum_t\mathbb{E}_{(\textbf{s}_t,\textbf{a}_t)\sim\rho}[r(\textbf{s}_t,\textbf{a}_t)+\alpha\mathcal{H}(\pi(\cdot\mid\textbf{s}_t))]$. However, it does not alter the fundamental optimization mechanism for the entropy term. The distribution shown in Fig. 2(b) is just one possible outcome of increasing the entropy of the discrete action distribution to a specified value. There is an infinite number of potential distributions with the same entropy value, and these distributions may have different probability orders.
>
> > "why each of the steps 1. to 4. in Section 3.2 is used"
>
> Zeroing out the maximum logits value is intended to simplify the calculation and visualization of the adaptive temperature effect (as shown in Figure 9 and 10 in the revised version). Adding or subtracting a constant value to all logits does not affect the probabilities computed by the softmax function. Analogous to continuous action spaces, our method establishes a lower bound for exploration in discrete spaces based on goal achievement while preserving the order of action probabilities. To further validate our approach, **we included a new baseline with label smoothing to guide exploration in the MiniGrid experiments**. The poor performance of this baseline supports our hypothesis that label smoothing leads to improper probability assignments, undermining exploration.
>
> > "there are methods designed specifically to address this problem' (local optima of intrinsic rewards) `These kind of methods need to be added as baselines"
>
> We agree that many methods were developed to address the issue of premature convergence caused by intrinsic reward methods, such as the Noisy-TV problem. Recent examples include NovelD, EIPO (suggested by the reviewer), DEIR, and others. In our work, we selected DEIR as the primary baseline because it demonstrated superior performance over several intrinsic reward methods, including NovelD, RND, ICM, and NGU, in the targeted experimental environments. In contrast, EIPO was only compared with RND in its evaluation. Additionally, we included comparisons with other popular baselines in this domain, such as ICM and RND. While we understand the importance of including more baselines, we kindly ask the reviewer to consider the resource constraints associated with a single work. Furthermore, our method is not designed to replace existing exploration approaches but to complement them. The core contribution of our work lies in the adaptive exploration lower bound based on the agent's performance, which can be integrated with other exploration methods. It is not intended as a standalone exploration strategy during training.
>
> > "pre-defined entropy schedules: linearly descreasing entropy, constant entropy, constant + linearly decreasing etc"
>
> **We included additional experiments in Appendix B.1** for the continuous control task Dog Balance Beam, using linearly decreasing and constant standard deviation schedules. As shown in the results, only the agent with a linearly decreasing standard deviation, similar to the curve discovered by our method, achieves performance comparable to GAGE. This finding further demonstrates the effectiveness of our approach. Predefined entropy schedules require extensive tuning of both the entropy values and the training duration, which can be computationally expensive. In contrast, our method introduces an adaptive schedule that significantly reduces this workload. We would greatly appreciate it if the reviewer could point us to specific papers focusing on predefined entropy schedules, as this would help us further refine our baselines and analyses.
>
>
> [Continued in third post due to character limit]

---

> ### Author Response · Authors · 2024-11-21
>
> > "Examples of continuous action benchmarks which require exploration such as AntMaze"
>
> We thank the reviewer for acknowledging the value of our proposed benchmarks. Regarding the hierarchical reinforcement learning (HRL) tasks suggested by the reviewer, we believe they are not directly suitable for evaluating our method, as our approach does not fall within the scope of HRL. For example, the AntMaze environment can be effectively treated as a combination of a high-level navigation task and a low-level locomotion task. Since we already demonstrated the effectiveness of our method in locomotion tasks (IsaacLab) and navigation tasks (MiniGrid), we believe our approach can also improve HRL algorithms when applied to sub-tasks at different levels.
>
> > "what is the relationship of the proposed approach compared to other methods that do not use intrinsic rewards"
>
> We thank the reviewer for recognizing the value of the proposed ''Game Consol'' problem. The MiniGrid environments are known for their extremely challenging exploration tasks, and they are frequently used by researchers to develop intrinsic reward or curriculum learning methods. Our work specifically aims to address premature convergence, and the ''Game Consol'' problem was proposed to highlight the challenges introduced by intrinsic rewards. We would greatly appreciate it if the reviewer could explicitly suggest other methods to consider for comparison, as this would help us further contextualize our approach."
>
> > "In ``Game Console'' type of problems, mostly intrinsic rewards cause problems?"
>
> We believe the issue is primarily caused by intrinsic rewards. Similar to the Noisy-TV problem, the agent becomes distracted by the novelty of the controllable aspects of the environment, even though these do not provide extrinsic rewards. However, the "Game Console" problem differs from the Noisy-TV problem, which can be addressed by algorithms like DEIR [1] that aim to identify the causal relationship between actions and observations. In contrast, algorithms such as DEIR would still struggle with the Game Console problem, as they remain susceptible to being trapped by controllable distractions.
>
> > "Please explain "More severely, for discrete actions, the entropy loss can not maintain the distribution shape"
>
> As shown in Fig. 2, when smoothing the original discrete distribution to a certain level (e.g., achieving a specific entropy value), methods like label smoothing and action smoothing provide a single definitive solution that preserves the original order of probabilities.
> In contrast, regularization using the entropy loss term focuses solely on achieving a target entropy value. Since there are infinitely many distributions with the same entropy, this approach may result in different probability orders, for example, by elevating the probability of the least promising action to the highest.
>
> > "in Equation 4, it seems that identical variances for all action dimensions is assumed?"
>
> Yes, we use identical lower bounds for variances across all action dimensions. While independently adapting the variances for different action terms might improve effectiveness, determining the optimal exploration-performance relationship for each action term is a non-trivial challenge. This could be an interesting future work direction.
>
> > "please define what entropy maximization means"
>
> **We modified the caption and notation in Fig. 2 to clarify the meaning of entropy maximization.** Here, entropy maximization refers to increasing the distribution's entropy, as is commonly done in reinforcement learning algorithms like SAC and PPO. It does not refer to calculating the maximum possible entropy of the original distribution.
>
> > "typos such as ``probablities''"
>
> We corrected the mentioned typos.
>
> We hope these responses help clarify the points the reviewer raised. Please don’t hesitate to reach out if there are any questions.
>
>
> [1] Shanchuan Wan *et al.*, DEIR: Efficient and Robust Exploration through Discriminative-Model-Based Episodic Intrinsic Rewards.

---

> > ### Comment · Reviewer_dGGt · 2024-12-02
> > **Remarks/questions**
> >
> > > Since there are infinitely many distributions with the same entropy, this approach may result in different probability orders, for example, by elevating the probability of the least promising action to the highest.
> >
> > I don't see how this could happen. The entropy bonus pushes the probabilities towards a uniform distribution. That means that it is not possible that an entropy bonus makes the probability of the least promising action the highest, or, even changes the order of the action probabilities.
> >
> > Assuming that entropy means here Shannon entropy which is a concave function and assuming we are trying to optimize a function of the form L = J + H, where J is the reward term and H is the entropy based term, then H does not change the order of probabilities in J when you try to maximize L.
> >
> > Can you provide a simple example where this reordering of action probabilities could happen with concrete real values that can be tested with pen and paper?
> >
> >
> >
> > > We included additional experiments in Appendix B.1 for the continuous control task Dog Balance Beam, using linearly decreasing and constant standard deviation schedules. As shown in the results, only the agent with a linearly decreasing standard deviation, similar to the curve discovered by our method, achieves performance comparable to GAGE. This finding further demonstrates the effectiveness of our approach. Predefined entropy schedules require extensive tuning of both the entropy values and the training duration, which can be computationally expensive. In contrast, our method introduces an adaptive schedule that significantly reduces this workload. We would greatly appreciate it if the reviewer could point us to specific papers focusing on predefined entropy schedules, as this would help us further refine our baselines and analyses.
> >
> > Thanks! This is a good starting point but should be done for all benchmarks. Since a predefined schedule is an obvious way of controlling randomness of a policy papers usually do not emphasize it. As examples, in [1], the entropy is kept constant (minimum entropy but in practice results in constant entropy) and in [2], a linearly decreasing predefined entropy schedule is used.
> >
> > [1] Haarnoja, Tuomas, Aurick Zhou, Kristian Hartikainen, George Tucker, Sehoon Ha, Jie Tan, Vikash Kumar et al. "Soft actor-critic algorithms and applications." arXiv preprint arXiv:1812.05905 (2018).
> >
> > [2] Pajarinen, Joni, Hong Linh Thai, Riad Akrour, Jan Peters, and Gerhard Neumann. "Compatible natural gradient policy search." Machine Learning 108 (2019): 1443-1466.

---

> ### Author Response · Authors · 2024-12-03
> **Why entropy maximization may alter the probability order of actions in a discrete distribution**
>
> We appreciate the reviewer’s continued engagement and acceptance of most of our explanations. However, we have noticed some remaining misunderstandings regarding why entropy maximization may alter the probability order of actions in a discrete distribution. Below, we address the remaining concerns raised by the reviewer:
>
> > "I don't see how this could happen. ... Can you provide a simple example where this reordering of action probabilities could happen with concrete real values that can be tested with pen and paper?"
>
> Entropy maximization in reinforcement learning typically increases the randomness of the action distribution by encouraging the probabilities of less likely actions to rise, aiming for a more uniform distribution. However, this process does not preserve the original order of action probabilities (i.e., their relative ranking). Here's why:
>
> 1. **Mathematical Independence of Entropy and Probability Order.**
> Entropy measures the overall randomness of a distribution. Mathematically, many distributions can share the same entropy value, even if their arrangements of probabilities differ. When maximizing entropy towards a uniform distribution, the optimization algorithm does not inherently constrain the order of probabilities. This means that actions with lower original probabilities can end up with higher probabilities after entropy regularization.
>
> 2. **Example in Practice.**
> To illustrate, we refer to the example in Figure 2, where three different techniques are used to flatten the original discrete action distribution ( 0.599 , 0.3 , 0.1 , 0.001 ). After flattening, the entropy is increased from 0.9 to 1.3 nats across all three techniques:
> - Label smoothing: This is achieved by $p'_i=(1-\epsilon)p_i+\epsilon\cdot\frac{1}{4}$ with $\epsilon=0.58$.
> - Action smoothing with softmax temperature: This is achieved by $p'_i=\text{softmax}(\frac{\ln{p_i}}{\tau})$ with $\tau=5.56$.
>
> Both techniques yield a single resulting distribution, as shown in Figures 2(c,d), because any different $\epsilon$ or $\tau$ value would result in a different entropy value.
> - Entropy regularization: In contrast, there are no explicit update rules, as in label smoothing or action smoothing. Mathematically, entropy regularization can produce any distribution with the desired entropy value. For example, besides the results in Figure 2(b,c,d), distributions such as (0.1,0.3,0.3,0.3), (0.19,0.15,0.26,0.4), (0.11,0.3,0.25,0.34), and many others are all valid results with the same entropy of 1.3 nats.
> 3. **Entropy Regularization in Reinforcement Learning.**
> In RL with an entropy maximization objective, entropy is increased by adjusting the policy network parameters $\theta$ through stochastic gradient descent (SGD) $\theta'=\theta+\eta\nabla_\theta \mathcal{H}$, where $\eta$ denotes the learning rate.
> However, this approach does not preserve the original order of $p_\theta(s)$ for the following reasons:
>     - **Non-concavity of $\mathcal{H}(\theta)$**. While Shannon entropy $\mathcal{H}(p_i)$ is a concave function, $\mathcal{H}(\theta)$, as a function of the policy network parameters $\theta$, becomes non-concave due to the non-linearities introduced by the neural network.
>     - **Overshooting in SGD**. SGD cannot guarantee monotonic increase in entropy. When the learning rate or the gradient magnitude is too large, overshooting can occur, leading to updates that fail to consistently increase entropy.
>     - **Lack of alignment between gradients**. The relationship between $\nabla_\theta \mathcal{H}$ and $\nabla_\theta p_i$ depends on the learnable network parameters. Even when $\mathcal{H(\theta')}>\mathcal{H(\theta)}$, there is no guarantee that $\frac{1}{K}>p_i(\theta'\mid s)>p_i(\theta\mid s)$ or $\frac{1}{K}<p_i(\theta'\mid s)<p_i(\theta\mid s)$, where $K$ represents action dimension. Moreover, there is no guarantee that $p_i(\theta'\mid s)>p_j(\theta'\mid s)$ given $p_i(\theta\mid s)>p_j(\theta\mid s)$. As a result, during each SGD update, the probability of an individual action may approach, deviate from, or even overshoot $\frac{1}{K}$, the probability of a uniform distribution.
> This lack of constraint allows the possible reordering of action probabilities during optimization.
>
> We hope this detailed explanation, along with the example and references to Figure 2, clarifies the distinction and addresses the reviewer’s concern. Please let us know if further clarification is needed.
>
> > "This is a good starting point but should be done for all benchmarks."
>
> We would like to include the results in the camera-ready version.

---

### Author Response · Authors · 2024-11-20
**General Response to Reviewer Feedback and Paper Changelog**

We thank the reviewers for their thoughtful feedback. We hope that our response below addresses all concerns raised, and we are excited about the improvements to our paper based on the great and helpful comments.

Here, we list major changes (marked in red in the pdf, the colors will be removed in the camera-ready version) we made according to the feedback:
1. We added an explanation of GAGE in terms of exploration. We want to clarify two concerns raised:
    - GAGE can also work without prior knowledge of the reward function. To demonstrate that, we added experiments on Humanoid locomotion task using standard PPO's episode return as the goal, and our results show that by simply using the reward sum instead of individual goal rewards, GAGE can already improve performance.
    - GAGE does not replace the intrinsic reward for tasks with sparse reward. Instead, it serves as a supplement to help the intrinsic reward techniques better explore the tasks, avoiding Noisy TV or even Game Console problems.
2. We added Label smoothing together with DEIR as a new baseline method in Minigrid experiments.
3. We added a stress test of different target speeds for the Humanoid task to show the robustness of our method.
4. We added extra experiments on the task Dog Balance Beam with different entropy schedules as baselines, as proposed by one reviewer.
5. We added a baseline with RND in all continuous tasks and conducted a hyperparameter tuning for RND on the Dog Balance Beam task.
6. We improved writing quality by revising Section 4.2 and fixing typos.
7. We increased the line width and legend font size in Figure 3 and reduced the spacing between plots in Figure 4(c).
8. To improve clarity, we renamed the variations of our methods from GAGE-50, GAGE-75, and GAGE-100 to GAGE-0.5, GAGE-0.75, and GAGE-1.0. We added an explanation of their differences to Section 4.1.
9. We corrected the results from GAGE-100 in the Humanoid Dribbling task. We are sorry about this, but it does not affect our experiment results.

---

### Meta-Review · Area_Chair_Vd2s · 2024-12-24

**Metareview:**

This paper introduces an adaptive temperature mechanism based on the agent’s performance relative to maximum performance. The stated goal of the mechanism is to avoid local optimum during optimization. Experiments are conducted to highlight that the agent does find some better optimum. Unfortunately, the paper is a bit too unrefined, particularly in sections 3 and 4, which leads to a lack of clarity on the method, its limitations, and questionable experiment results. The paper is also missing a citation for an extremely similar method proposed by Gullapalli (1992). See the comments below for more details.


The background section for this paper is long, and it does not seem to build up to properly frame the contribution. The primary content of the paper starts halfway down page 4. It could be beneficial to get to this point faster.

Figure 2: it is difficult to understand what this figure is contributing without providing the equations that would produce these plots.

What impact does the forced lower bound have of optimization? This feature is introduced, but it is never demonstrated how this impacts the algorithm or that it is necessary.

For the experiments in Figure 3, plotting the median return with 25%-75% quantiles is not a clear choice. This work aims to prevent the agent from getting trapped in a local optimum. These plots could only indicate if the agent didn’t get stuck in local optima 75% of the time. There will also be significant uncertainty with where these quantiles are with only 10 seeds.

There is no accounting for hyperparameter tuning in comparing the success of each method. It has been shown that large step sizes (Jordan et al. 2024) lead policy gradient methods to getting trapped in plateaus. Without considering the impact of the step size from these experiments, it is impossible to understand if the method worked as intended or if there is some other confounding factor. Plus, with further hyperparameter tuning, PPO could get to the same performance level without this trick. If the only gauge of success of this method is getting good performance with a specific hyperparameter setting, how do we know it is doing anything of value?


Where is the evidence that the tasks in Figure 3 are hard exploration problems? These may be challenging optimization problems, but whether they are hard exploration problems is unclear. In fact, I see no reason why this method of exploration could solve hard exploration problems efficiently. The primary exploration mechanism is random sampling, which is widely known to not solve hard exploration problems efficiently. Furthermore, it could be the case that having high entropy action distributions could make the agent get stuck in a local optimum that prefers having lots of noise. The experiments all try to show that the method is universally applicable, but this is not going to be the case. Every method has a limitation, and it should be made clear. The method and claims of this paper need to be adjusted to scope and clarify the applicability of this method correctly.

The experiment of robustness to reward shaping is very interesting! However, it is unclear why this method would have any impact on this robust. Since it is included in the main paper, I would expect further investigation. Revealing this connection could produce further insights on the method and how it impacts policy optimization.

I think the method is interesting and could be very useful for the community to understand its value. However, I do not think this is accomplished in the paper’s current form.


Gullapalli, Vijaykumar. "A stochastic reinforcement learning algorithm for learning real-valued functions." Neural networks 3.6 (1990): 671-692.

Scott M Jordan, Samuel Neumann, James E Kostas, Adam White, and Philip S Thomas. "The Cliff of Overcommitment with Policy Gradient Step Sizes." Reinforcement Learning Journal, vol. 2, 2024, pp. 864–883.

**Additional Comments On Reviewer Discussion:**

The reviewers and authors had some discussion, but it did not seem to sway any reviewer strongly.

---

### Decision · Program_Chairs · 2025-01-22

Reject